# Study: International Multicentric Minimally Invasive Liver Resection for Colorectal Liver Metastases (SIMMILR-CRLM)

**DOI:** 10.3390/cancers14061379

**Published:** 2022-03-08

**Authors:** Andrew A. Gumbs, Eric Lorenz, Tzu-Jung Tsai, Lee Starker, Joe Flanagan, Andrea Benedetti Cacciaguerra, Ng Jing Yu, Melinda Bajul, Elie Chouillard, Roland Croner, Mohammad Abu Hilal

**Affiliations:** 1Departement de Chirurgie Digestive, Centre Hospitalier Intercommunal de Poissy/Saint-Germain-en-Laye 10, Rue du Champ Gaillard, 78300 Poissy, France; melinda.bajulidei@gmail.com (M.B.); chouillard@yahoo.com (E.C.); 2Department of General-, Visceral-, Vascular- and Transplantation Surgery, University of Magdeburg, Haus 60a, Leipziger Str. 44, 39120 Magdeburg, Germany; eric.lorenz@med.ovgu.de (E.L.); roland.croner@med.ovgu.de (R.C.); 3Department of Surgery, Koo Foundation Sun Yat-Sen Cancer Centre, Taipei 112019, Taiwan; gratiatsai@gmail.com (T.-J.T.); jingyu_ng@nuhs.edu.sg (N.J.Y.); 4Morristown Medical Center, Department of Surgical Oncology, Morristown, NJ 07960, USA; lstarker7@gmail.com (L.S.); joeflanagan1016@gmail.com (J.F.); 5Unità Chirurgia Epatobiliopancreatica, Robotica e Mininvasiva, Fondazione Poliambulanza Istituto Ospedaliero, Via Bissolati 57, 25124 Brescia, Italy; dott.benedetti@gmail.com (A.B.C.); abuhilal9@gmail.com (M.A.H.)

**Keywords:** hepatectomy, laparoscopic, robotic, minimally invasive, liver resection, colorectal liver metastasis, multicentric, international, laparoscopy, robot-assisted

## Abstract

**Simple Summary:**

This is study of an international multicentric cohort after minimally invasive liver resection (SIMMILR) from six international centers evaluating short-term outcomes after minimally invasive liver resection for patients with three or fewer colorectal liver metastases that measure less than or equal to 3 cm, or a solitary tumor less than or equal to 5 cm (Milan Criteria). Propensity score matching was done to reduce bias. Comparisons were done between open, laparoscopic and robotic liver resections. Laparoscopic and robotic approaches may have short-term benefits when compared to open hepatectomy. Future studies will include an analysis of overall and recurrence-free survival curves by stage and type of neoadjuvant treatments received.

**Abstract:**

(1) Background: Here we report on a retrospective study of an international multicentric cohort after minimally invasive liver resection (SIMMILR) of colorectal liver metastases (CRLM) from six centers. (2) Methods: Resections were divided by the approach used: open liver resection (OLR), laparoscopic liver resection (LLR) and robotic liver resection (RLR). Patients with macrovascular invasion, more than three metastases measuring more than 3 cm or a solitary metastasis more than 5 cm were excluded, and any remaining heterogeneity found was further analyzed after propensity score matching (PSM) to decrease any potential bias. (3) Results: Prior to matching, 566 patients underwent OLR, 462 LLR and 36 RLR for CRLM. After PSM, 142 patients were in each group of the OLR vs. LLR group and 22 in the OLR vs. RLR and 21 in the LLR vs. RLR groups. Blood loss, hospital stay, and morbidity rates were all highly statistically significantly increased in the OLR compared to the LLR group, 636 mL vs. 353 mL, 9 vs. 5 days and 25% vs. 6%, respectively (*p* < 0.001). Only blood loss was significantly decreased when RLR was compared to OLR and LLR, 250 mL vs. 597 mL, and 224 mL vs. 778 mL, *p* < 0.008 and *p* < 0.04, respectively. (4) Conclusions: SIMMILR indicates that minimally invasive approaches for CRLM that follow the Milan criteria may have short term advantages. Notably, larger studies with long-term follow-up comparing robotic resections to both OLR and LLR are still needed.

## 1. Introduction

Gagner performed the first laparoscopic liver resection in 1992 [1]. Progressively, laparoscopy has become the gold standard for minor hepatectomies, mainly left-sided [2,3]. Due to the absence of regimented training programs, pioneering hepatic-pancreatic and biliary (HPB) surgeons had to teach themselves minimally invasive surgical techniques (MIS) [4,5,6]. As experience grew, early adopters of MIS began performing more major hepatectomies, including those with resection of neighboring organs and complex biliary reconstructions [7,8,9].

Having emerged in the late 2000s, the new generation of fellowship-trained (FT) surgeons benefitted from the programs knitted by the pioneers experience [10,11,12,13]. This apprenticeship system was formalized in 2007 by the creation of the first International Hepatic-Pancreatic-Biliary Association (IHPBA) sanctioned MIS HPB fellowship. Ever since, increasing numbers of fellowship trained HPB surgeons have incorporated MIS into their curriculum (https://www.ihpba.org/27_HPB-Fellowship-Registry.html, accessed on 22 December 2021) [12]. The fellowship era changed the indications for MIS liver resection. From the initial lesion less than 5 cm and located in the anterior segments, pioneering centers quickly extended their indications to the posterior segments [14,15,16]. Progressively, major, central, and extended hepatectomies were reported.

More recently, complete robotic systems have evolved, and increasing numbers of liver resection are being done with robotic assistance [17,18,19]. Because of these changing habits we wanted to look at short- and long-term outcomes after liver resection to see what differences remain between open, laparoscopic and robotic approaches. To our knowledge there are two randomized controlled trials comparing open and laparoscopic hepatectomy for colorectal liver metastases [20,21]. There is a retrospective, multicentric international study that compares laparoscopic and robotic resection techniques for metastatic colorectal cancer [22]. However, there is no multicentric international study looking at three different resection techniques such as open liver resection (OLR), laparoscopic liver resection (LLR) and robotic liver resection (RLR). We wanted to do this study to give a broader indication of how liver surgeons are managing CRLM around the world.

## 2. Materials and Methods

### 2.1. Patient Selection & Indication for Surgery

We collected perioperative data as well as follow-up data of all eligible patients who underwent OLR, LLR or RLR for colorectal metastasis between June 2004 and October 2020. All hepatectomies for colorectal liver metastases done by single surgeon with experience in both open and minimally invasive liver resection at six different international centers (two in USA and one each in Italy, France, Germany and Taiwan) were divided into open, laparoscopic and robotic cohorts and compared. Minimum case requirements for participating HPB surgeons in this study was the completion of at least, 50 laparoscopic and/or robotic hepatectomies and, at least, 40 hepatectomies for CRLM. The six centers consisted of two pioneers, two early adopters and two fellowship-trained (FT) hepatobiliary surgeons to give a balanced picture of what is currently occurring internationally for patients with resectable CRLM. Although all surgeons did open and laparoscopic hepatectomy, only three also did hepatectomies with use of the complete robotic surgical system (two early adopters and one fellowship-trained).

The term robotic was reserved for procedures done with a complete robotic surgical system such as the da Vinci (Intuitive Surgical, Inc., Sunnyvale, CA, USA) or Versius robots (Versius Robotics, CMR, Cambridge, UK). Indications for surgery were resectable synchronous or metachronous liver metastases in patients with colorectal cancer. Prior to surgery we obtained histological confirmation if necessary. Otherwise, clear evidence for liver metastasis in imaging was sufficient. However, for the purpose of a patient-centered, tailored treatment approach perioperative/operative/alternative treatment options were discussed in a multidisciplinary tumor board (MDT) prior to surgery. Absolute contra-indications to minimally invasive liver surgery (MILS) included closed angle glaucoma and intracranial hypertension. Severe lung disease was considered to be a relative contraindication.

### 2.2. Study Endpoints

We performed a retrospective review of all patients who underwent liver resection for colorectal metastases. For the PSM we excluded patients with more than three metastases measuring ≥3 cm, solitary metastasis >5 cm and evidence of macrovascular invasion. Although originally devised to help in treating patients with hepatocellular cancer, the Milan criteria were also used to identify patients with CRLM that would benefit from liver resection and because of this we decided to use it as an exclusion criteria [23]. This was also done to reduce the perceived bias that “easier” tumors are removed minimally invasively when compared to open approaches. Patients who had undergone associating liver partition and portal vein ligation for staged hepatectomy (ALLPS), previous ablation and repeat liver resections were excluded. The files of patients started with either laparoscopy or robotic assistance were analyzed on an intention to treat basis. The data that support the findings of this study are available from the corresponding author upon reasonable request.

For further comparative analysis patients were divided into three groups (OLR, LLR and RLR) depending on the surgical technique used for resection. The primary endpoint of the study was postoperative short-term mortality (death within 30 and 90 days). Secondary endpoints were intraoperative parameters (blood loss, OR time), length of hospital stay, complete oncologic resection and severe postoperative complications. The Dindo-Clavien classification system was used in case of postoperative complications with major complications defined as greater than grade 2 [24]. Finally, propensity score matching was also done to get a more accurate comparison between techniques. Written informed consent was obtained on all patients. On the informed consent that is signed by each patient, it is explained that their anonymous patient data may be used to perform future studies. The STROBE statement checklist for reporting of observational studies was used during the drafting and editing of the manuscript.

### 2.3. Surgical Techniques

LLR was performed as total laparoscopic procedure. Trocars were placed depending on the site of liver resection as described previously [25]. Laparoscopic minor hepatectomy was usually performed via three to four trocars. Major and extended laparoscopic hepatectomies were performed with five to six trocars, usually < 1 hand-breadth below the right costal margin. For parenchymal dissection, we used ultrasonic shears, a harmonic scalpel and a laparoscopic CUSA (cavitron ultrasonic surgical aspirator) or an aquajet. For RLR, different versions of the DaVinci System (Intuitive, Santa Clara, CA, USA, used versions: Si, X, Xi) were used depending on the center. The resection techniques used were performed as described previously [26]. Open liver resection was usually done via a right sub-costal incision, extended to a chevron incision for major and extended major hepatectomies. In LLR the laparoscopic vascular linear gastrointestinal anastomosis stapler was used for vascular transection. In RLR vessels were transected with vessel sealer, Hem-O-lok clip applier and scissors. If the Pringle was used even once, the technique was considered to have been utilized.

### 2.4. Definition of Extent of Liver Resection

Preoperative embolization was obtained when an extended major hepatectomy was planned. Minor hepatectomy was defined as less than three hepatic segments, major hepatectomy was defined as three or more hepatic segments, Extended hepatectomy was defined as resection of more than four hepatic segments. Central hepatectomy was defined as removal of hepatic segments 4, 5 and 8. Lesions in the deep or deeper segments were defined as metastases in segments 4B, 7 or 8.

### 2.5. Data Analysis

Statistical data analysis was performed using the Social Science Statistics software (www.socscistatistics.com, accessed on 22 December 2021) and SPSS (version 26; IBM, Armonk, NY, USA). Categorical data (nominal/ordinal) are presented as absolute (n) and/or relative values (%). Differences between the groups were tested using the Pearson’s χ^2^ or Fisher’s exact test (if at least one cell had a cell count of less than 5). Continuous data was expressed as mean (SD—standard deviation). Differences between continuous variables were analyzed using the Mann–Whitney U-test for continuous variables < 200 distinct values or Student’s *t*-test for data sets with 200–500 distinct values. For all analyses, differences with a two-sided *p*-value < 0.05 were considered to be significant (no adjustment for multiplicity).

### 2.6. Propensity Score Matching

Propensity score was calculated using a logistic regression model in SPSS (version 26; IBM, Armonk, NY, USA). Potential confounding variables/predictors were chosen in agreement with all hepatopancreatobiliary (HPB) surgeons of the participating study centers. Those predictors comprise: age, gender, ASA (American Society of Anesthesiology) score, prevalence of previous abdominal surgery, prevalence of previous chemotherapy, prevalence of liver cirrhosis, extent of performed liver resection (minor vs. major resection), number of resected metastases, maximum size of resected metastasis. Metastasis size was based on final pathological result. We set matching tolerance to 0.2. Surgical approach (open vs. laparoscopic vs. robotic) was used as dependent group variable. Matching was performed in a 1:1 ratio of nearest neighbor without replacements. Patients receiving open liver resection (OLR) were matched to patients receiving laparoscopic liver resection (LLR) as well as patients receiving robotic liver resection (RLR). Additionally, LLR patients were matched to RLR patients. We excluded patients with incomplete information on predictors and patients without matching from PSM.3.

## 3. Results

From June 2004 until October 2020, a total of 1064 hepatectomies were performed by six surgeons. A total of 566 patients (53.2%) were included in the open hepatectomy group, 462 patients (43.4%) in the laparoscopic group and 36 in the robotic group (3.4%) (Table 1). The percentage of cases by center done minimally invasively (laparoscopic and/or robotic) ranged from 13.5–87.2% prior to PSM. After matching, the percentage of liver resections done via an open, laparoscopic or robotic approach by center was: 45.4/69.3/0 (center 1), 36.8/3.1/39.6 (center 3), 0/12.9/0 (center 4), 6.1/1.8/30.2 (center 5) and 11.7/12.9/30.2 (center 6), respectively. No cases from center 2 were included after matching. Figure 1 shows the percentages of approaches (OLR, LLR, RLR) used for the five centers included in the PSM over the period of the study (2004–2020).

When unmatched patients undergoing OLR were compared to LLR, there were no significant differences between age, sex distribution, BMI and carcinoembryonic antigen (CEA) levels. However, increases in ASA scores were statistically significant (*p* = 0.03) and an increased history of previous abdominal surgery, history of neoadjuvant chemotherapy, mean metastasis size and number of metastases were highly statistically significant. The rate of cirrhosis was highly significantly more prevalent in the LLR group compared to the OLR group. Cirrhosis was due to either steato-hepatitis or alcohol abuse. No patients had viral hepatitis and all cirrhotics were Child’s A. Lastly, lesions were significantly more prevalent in the deeper segments in the OLR group, *p* = 0.03. No differences were noted in the utilization of the Pringle maneuver.

When unmatched patients who had undergone OLR were compared to the RLR, ASA class and presence of cirrhosis were significantly higher in the RLR group, *p* = 0.02 and 0.03, respectively. However, significantly more patients received neoadjuvant chemotherapy, had undergone previous surgery and had on average higher preoperative CEA serum levels, *p* = 0.03, 0.04 and 0.08, respectively. Furthermore, more and larger metastases were removed in the OLR group: *p* = 0.003 and 0.007, respectively. Although lesions tended to be in the deeper segments more often in the OLR group, this did not attain statistical significance, *p* = 0.06. The Pringle maneuver was used significantly more frequently in the OLR group when compared to the RLR group, *p* = 0.004).

When unmatched patients who had undergone LLR were compared to RLR, the ASA class was statistically higher in the RLR group, *p* = 0.001, and the Pringle maneuver was used statistically more often in the LLR group, *p* = 0.02.

### 3.1. Differences between Open (OLR) and Laparoscopic Liver Resection (LLR)

The percentages Table 2 describes and compares demographics, confounding variables and clinical outcome variables of patients undergoing OLR and LLR for colorectal liver metastases before and after propensity score matching (PSM). During PSM 142 patients were matched 1:1 in each group. After matching cirrhosis and number of resected metastasess were no longer significantly different. Both before and after PSM, blood loss, hospital stay and Clavien–Dindo ≥ grade 3 complication rates were all highly statistically significantly greater after OLR when compared to LLR. Notably after PSM, R0 resection and 30-day and 90-day mortality rates were not statistically different. All major complications involved either percutaneous drainage of bile leaks or abscesses and/or need for endoscopic biliary drainage. No reoperations were noted.

### 3.2. Differences between Open (OLR) and Robotic Liver Resection (RLR)

Table 3 describes and compares demographics, confounding variables and clinical outcome variables of patients undergoing OLR and RLR for colorectal liver metastases that follow the Milan criteria before and after propensity score matching (PSM). During PSM, 22 patients were matched 1:1 in each group. After PSM no differences were noted in average metastasis size, number of resected metastatic lesions, percentages of major hepatectomies, location in the deep segments or utilization of the Pringle maneuver between the two approaches. Although before matching patients in the RLR group had a significantly higher prevalence of liver cirrhosis (10.7% vs. 0.3%; *p* < 0.001) and severe comorbidities indicated by significantly higher mean ASA score (2.5 vs. 2.1; *p* < 0.001); and metastases were found significantly more often in the deep segments (44.0% vs. 35.7%; *p* = 0.04) and the Pringle maneuver was used more frequently in the OLR group (25% vs. 3.6%; *p* = 0.009), these differences were insignificant after the matching process. Notably, after PSM estimated blood loss remained significantly lower in cases of RLR when compared to OLR (596.8 mL vs. 250.0 mL; *p* < 0.008). All major complications involved either percutaneous drainage of bile leaks or abscesses and/or need for endoscopic biliary drainage. No reoperations or mortalities were noted.

### 3.3. Differences between Laparoscopic (LLR) and Robotic Liver Resection (RLR)

Table 4 describes demographics and clinical outcome variables of LLR and RLR. Although before matching patients in the RLR group had a significantly higher number of metastases resected (1.4 vs.1.0; *p* = 0.003); and significantly more major resections (47.5% vs. 17.9%; *p* = 0.002) and the Pringle maneuver (28.0% vs. 3.6%; *p* = 0.003) was used more frequently in the LLR group, these differences were insignificant after the matching process. During PSM, 21 patients were matched 1:1 in each group. Notably, estimated blood loss was significantly less in RLR (219.2 mL vs. LLR group 408.6 mL; *p* < 0.001) before PSM and remained less in the RLR group after PSM (RLR 223.7 mL vs. LLR 777.7 mL; *p* = 0.04). One patient undergoing LLR had a bile leak that had to be drained. No reoperations were required and there were no mortalities.

## 4. Discussion

When the study started, the aim was to compare MIS liver resections to open resections. However, because of the vast discrepancy in utilization of MIS techniques for hepatectomy (laparoscopic vs. robotic) and the fact that, aside from cirrhotics, sicker, more complex patients with larger and more tumors are being approached via open approaches (Table 1), we attempted a propensity matching score of patients with CRLM following the Milan criteria to get a more accurate and honest comparison [27,28,29,30,31,32].

LLR seems to have benefits in improved lengths of hospitalization and decreased complication rates, when compared with the OLR. Notably, estimated blood loss is decreased after both LLR and RLR when compared to OLR after PSM. Lastly, operative times are not statistically different when OLR is compared to either minimally invasive modality (Table 4). Regardless of the modality used, OS and RFS are similar and not statistically different. Although RFS tended to be decreased after RLR when compared to either OLR or LLR this was not statistically significant (*p* = 0.06 and 0.08, respectively). Additionally, the small numbers in each group and fact that OS is not significantly different make it unlikely that this difference would be found in larger series.

Abu Hilal et al. recently published a meta-analysis comparing the results of open to laparoscopic liver resection. A total of 917 patients were identified with 427 in the laparoscopic group and 490 in the open group. Out of 417 studies, only eight met criteria for analysis [33]. MIS was found to decrease LOS and morbidity rates in this meta-analysis [34]. The LOS ranged from 6 to 12 days in the MIS patients compared to 8 to 16 days with an average of 3.09 days less in the MIS cohorts (CI = −4.96; −1.22) [33]. As mentioned in our study, LOS averaged 5.0 days in the laparoscopic group, six to seven days in the robotic group and eight to nine days in the open group, and this difference was only statistically significant when LLR was compared to OLR (Table 2). This is consistent with the largest meta-analysis comparing laparoscopic and open hepatectomies [33].

Clavien–Dindo complications of grade 3 and above occurred in 4–6% of the laparoscopic patients, 0–22% in the robotic cohort and 25–35% in the open cohort and, although this difference was statistically significant only when the LLR was compared to the OLR group, these rates are well within the range published in the literature of 5–55% after laparoscopic hepatectomy, 17–20% after robotic resection and 17–61% after open hepatectomy [33,35,36]. The major cause of morbidity after major liver resection is bile leak with 13.5% of patients after major hepatectomy found to have a bile leak in one study [37]. When both minor and major liver resections are analyzed, the rate of biliary fistula was found to decrease to 1.5% in a large review of 2804 MIS hepatectomy patients published in 2009 [38].

All participants had to have done, at least 50 minimally invasive hepatectomies to be included in this trial because the range of the traditional learning curve has been described by Brown and Geller to range between 45 and 60 [39,40]. Estimation of the length of the learning curve emanated from open hepatic surgeons who then taught themselves MIS techniques, here called the pioneers, which is further discussed below [6,9,41]. It seems that the lack of previous extensive open hepatectomy experience could be successfully compensated by advanced MI HPB fellowship training [42]. Hand-assistance is particularly useful early in one’s experience to decrease the conversion rate. One of the benefits of this MI hepatectomy technique is that the robotic laparoscope holder (ViKY, Endocontrol, Grenoble, France) gives some of the advantages of robotics, but unlike full surgical systems, the small size of the laparoscope holder enable the primary surgeon to remain in contact with the patient and even use hand-assistance if necessary. Furthermore, unlike the full surgical systems, haptics can be maintained throughout the procedure by way of hand-held instruments.

In 2012 a group from Belgium reported a case-controlled study of 20 patients with colorectal liver metastasis undergoing laparoscopic major hepatic resection compared to 20 resected via an open approach [34]. Notably, the rate of R0 resection in our series is comparable to other published rates of 95% [34].

The open approach is reserved for combined cases involving dual surgeons when one surgeon is not comfortable with minimal invasion, or if patients have a previous history of an abdomen not amenable to laparoscopy. Even metastases as large as 17 cm have been removed with MIS in this series, as a result, size does not seem to be an absolute contraindication to minimal invasion. This is highlighted by the frequent use of MIS in massive splenomegaly [43].

Cost has often been considered as the Achilles heel of MIS. However, a paper from Emory University published in 2014 showed that although the intraoperative costs during MIS hepatectomy were greater than open operating room costs, the overall hospital costs for these two approaches were equivalent [44]. One of the main factors contributing to the costs was the increased operative time during MIS hepatectomy [44]. Consequently, one wonders if MIS hepatectomy may one day prove to be more cost-effective than open hepatectomy, especially as the newer generation of MI HPB FT surgeons reaches the post-learning-curve phase of mastery and as more studies show that the hospital stays is significantly shorter than after open resections.

An additional potential benefit of the minimally invasive approach to liver metastases could be in the decreased intra-abdominal scar formation after laparoscopy. The first study to analyze the feasibility of repeat laparoscopic hepatectomy came from the group at the Institut Mutualiste Montsouris in Paris [45]. Early reports indicate that this approach may have survival advantages, but also an increased risk of liver insufficiency [30,46]. In this study 11% (*n* = 10) patients underwent repeat laparoscopic hepatectomy and none had to be converted to an open approach. However, variability in tumor type and the small sample size make further analysis moot.

Although there is significant heterogeneity between the approaches used by center, the centers included in this study are referral centers and are highly experienced in minimally invasive surgery. As a result, the presented data might not be totally representative for real world situations. Another limitation of the study is the lack of longer follow-up and low numbers, particularly, in the robotic group. Additionally, the case number of RLR is small and the cases are from only three centers. This is due to the fact that robotic surgery of the liver is still in its infancy. We had strict criteria inclusion and exclusion criteria for our PSM and needed to include nonrobotic centers to get enough patients to make any meaningful comments on the various approaches to liver surgery. Before matching, Pringle maneuver was used significantly more during OLR and LLR when compared to RLR, after matching this difference was no longer statistically significant. The observation that patients had on average 540 mL more blood loss after LLR and RLR after matching is probably due to the fact that more patients had tended to have cirrhosis in the LLR arm.

In the future, as RLR becomes more common we believe that it will indeed be possible to do a better multicentric study where each center has enough open, laparoscopic and robotic liver resection, however, we are not there yet. Importantly, we believe that future liver surgeons should become versed in all three approaches and that future studies on liver surgery should include data from all three techniques and not solely be limited to binary analyses (i.e., OLR vs. RLR).

## 5. Conclusions

This study of international multicentric minimally invasive liver resection (SIMMILR) is a retrospective PSM study of CRLM, which reveals that LLR may result in decreased length of hospital stay and complication rates in the compared to OLR. Both minimally invasive approaches (laparoscopic and robotic) seem to have the added benefit of significantly decreased blood loss when compared to OLR without significantly increasing operative times. Minimally invasive liver resection may enable equivalent parenchymal sparing hepatectomies to OLR in metastases that follow the Milan criteria. Larger randomized-controlled trials recording stage at diagnosis or whether or not CRLM are synchronous or metachronous and type of neoadjuvant treatments given are still needed.

## Figures and Tables

**Figure 1 cancers-14-01379-f001:**
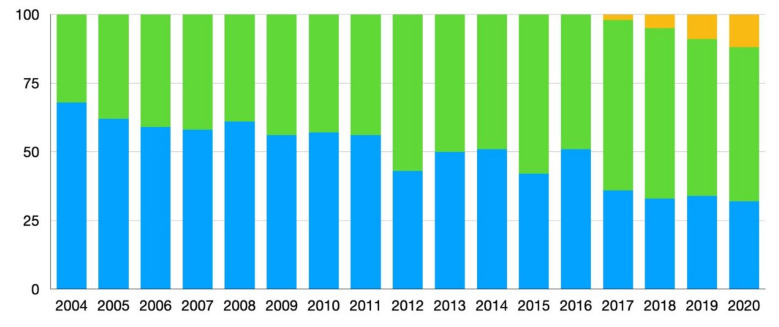
Changes in percentages of surgical approach used from 2004–2020 of all cases of liver resection for colorectal liver metastases (CRLM) after matching. Blue = open liver resection (OLR), green = laparoscopic liver resection (LLR), yellow = robotic liver resection (RLR).

**Table 1 cancers-14-01379-t001:** Demographics/confounding variables of patients receiving open (OLR), laparoscopic liver resection (LLR) or robotic liver resection (RLR) for colorectal liver metastasis before propensity score matching and before the exclusion criteria were applied (#=number of metastases).

Demographics	Open *n* = 566 (53.2%)	Open vs. Lap *p*-Value	Lap *n* = 462 (43.4%)	Lap vs. Robotic *p*-Value	Robot *n* = 36 (3.4%)	Robotic vs. Open *p*-Value
Mean Age (range)	63.7 (27–88)	0.3	63.9 (26–93)	0.2	62.0 (41–83)	0.4
M:F	286:280	0.09	259:204	0.9	21:15	
ASA (%)	1 = 88 (15.5) 2 = 328 (58.0) 3 = 146 (25.8) 4 = 4 (0.7)	0.03	1 = 102 (22.1) 2 = 258 (55.8) 3 = 102 (22.1) 4 = 0	0.001	1 = 1 (2.7) 2 = 19 (52.7) 3 = 16 (44.6) 4 = 0	0.02
BMI kg/m^2^ (range)	26.5 (15.5–47.4)	0.3	26.7 (17.4–43.8)	0.5	26.2 (19.4–39.0)	0.8
Neoadjuvant Chemotherapy (%)	479 (84.6)	<0.001	355 (76.8)	0.3	25 (69.4)	0.03
Previous Surgery	411 (72.6)	<0.001	261 (56.5)	1	20 (55.6%)	0.04
Cirrhosis (%)	3 (0.5)	<0.001	18 (3.9%)	0.2	3 (8.3%)	0.004
CEA (range)	47.6 (0.4–520.5)	0.8	51.6 (0.4–962)	0.1	29.3 (3.4–38)	0.08
PVE (%)	6 (1.1)	0.09	12 (2.2)	1	0 (0)	1
Metastasis Size (cm) (range)	4.1 (1.0–29.0)	<0.001	3.0 (0.1–17.0)	0.4	3.3 (0.6–9.0)	0.003
# (range)	2.5 (1–12)	<0.001	1.7 (1–8)	0.4	1.5 (1–6)	0.007
Location in Deep Segments (%)	269 (47.5)	0.03	189 (40.9)	0.3	11 (30.5)	0.06
Pringle Maneuver (%)	171 (30.2)	0.1	120 (26.0)	0.02	3 (8.3%)	0.004

# = number of metastases.

**Table 2 cancers-14-01379-t002:** Demographics/confounding variables and clinical outcome variables of patients receiving open (OLR) or laparoscopic liver resection (LLR) for colorectal liver metastasis before and after propensity score matching.

Demographics	Before Matching			After Matching		
	OLR *n* = 375	LLR *n* = 378	*p*-Value	OLR *n* = 142	LLR *n* = 142	*p*-Value
Confounding Variables						
Age, mean ± SD (years)	64.0 ± 10.7	63.8 ± 11.7	0.7	64.4 ± 10.8	64.5 ± 12.4	0.9
Gender; *n* (%)			0.4			0.6
Female	148 (39.5)	161 (42.6)		66 (46.5)	70 (49.3)	
Male	227 (60.5)	217 (57.4)		76 (53.5)	72 (50.7)	
ASA score, mean ± SD	2.1 ± 0.7	2.2 ± 0.6	0.08	2.2 ± 0.7	2.2 ± 0.5	0.7
Prior abdominal surgery, *n* (%)	215 (59.1)	222 (58.7)	0.9	91 (64.1)	87 (61.3)	0.6
Liver cirrhosis, *n* (%)	1 (0.3)	15 (4.2)	0.0001	0 (0)	0 (0)	
Prior chemotherapy, *n* (%)	274 (73.7)	261 (69.4)	0.2	104 (73.2)	99 (69.7)	0.5
Number of resected metastases, mean (range)	1.7 (1–5)	1.0 (1–3)	0.0001	1.0 (1–3)	1.0 (1–3)	1.0
Diameter of largest metastasis, mean ± SD (cm)	2.6 ± 1.4	2.4 ± 11.8	0.1	2.7 ± 1.30	2.5 ± 1.1	0.3
Extent of liver resection, *n* (%)			0.6			0.8
Major resection (≥3 segments)	144 (45.7)	177 (47.5)		12 (8.5)	11 (7.7)	
Minor resection	171 (54.3)	196 (52.5)		130 (91.5)	131 (92.3)	
Location in deep segments (%)	165 (44.0)	147 (38.3)	0.1	61 (42.9)	50 (35.2)	0.2
Pringle maneuver (%)	101 (26.9)	106 (28.0)	0.7	36 (25.4)	38 (26.7)	0.9
Outcome variables						
Measured blood loss, mean ± SD (mL)	706.9 ± 674.4	408.6 ± 484.2	0.0001	613.6 ± 606.8	340.8 ± 510.0	0.0001
Operative time, mean ± SD (min)	235.9 ± 105.8	246.5 ± 115.0	0.2	225.7 ± 107.6	217.0 ± 114.1	0.5
Postoperative hospital stay, mean ± SD (d)	9.6 ± 6.4	5.4 ± 4.7	0.0001	9.1 ± 6.0	5.0 ± 3.1	0.0001
Postoperative Clavien–Dindo morbidity ≥ grade 3, *n* (%)	115 (31.6)	35 (9.4)	0.0001	33 (23.9)	9 (6.4)	0.0001
30-day mortality, *n* (%)	2 (0.5)	0 (0)	0.0001	0 (0)	0 (0)	
90-day mortality, *n* (%)	4 (1.1)	4 (1.1)	1.0	0 (0)	0 (0)	
R1 resection, *n* (%)	51 (16.3)	27 (7.6)	0.001	16 (11.8)	12 (8.8)	0.4

**Table 3 cancers-14-01379-t003:** Demographics/confounding variables and clinical outcome variables of patients receiving open (OLR) or robotic liver resection (RLR) for colorectal liver metastasis before and after propensity score matching.

Demographics	Before Matching			After Matching		
	OLR *n* = 375	RLR *n* = 28	*p*-Value	OLR *n* = 22	RLR *n* = 22	*p*-Value
Confounding Variables						
Age, mean ± SD (years)	64.0 ± 10.8	61.8 ± 11.0	0.3	60.5 ± 10.4	60.4 ± 12.4	1.0
Gender, *n* (%)			0.03			1.0
Female	148 (39.5)	17 (60.7)		16 (72.7)	16 (72.7)	
Male	227 (60.5)	11 (39.3)		6 (27.3)	6 (27.3)	
ASA score, mean ± SD	2.1 ± 0.7	2.5 ± 0.6	0.0001	2.6 ± 0.5	2.6 ± 0.7	0.8
Prior abdominal surgery, *n* (%)	215 (59.1)	16 (57.1)	0.8	11 (50.0)	12 (54.5)	0.8
Liver cirrhosis, *n* (%)	1 (0.3)	3 (10.7)	0.001	0 (0)	0 (0)	
Prior chemotherapy, *n* (%)	274 (73.7)	19 (70.4)	0.7	18 (81.8)	19 (86.4)	1.0
Number of resected metastases, mean (range)	1.7 (1–5)	1.4 (1–3)	0.1	1.4 (1–3)	1.5 (1–3)	0.8
Diameter of largest metastasis, mean ± SD (cm)	2.6 ± 1.4	2.5 ± 1.1	0.8	2.7 ± 1.2	2.5 ± 1.1	0.6
Extent of liver resection, *n* (%)			0.004			1.0
Major resection (≥3 segments)	144 (45.7)	5 (17.9)		4 (18.2)	5 (22.7)	
Minor resection,	171 (54.3)	23 (82.1)		18 (81.8)	17 (77.3)	
Location in deep segments (%)	165 (44.0)	10 (35.7)	0.04	9 (40.9)	6 (27.2)	0.5
Pringle maneuver (%)	94 (25.1)	1 (3.6)	0.009	5 (22.7)	1 (4.5)	0.2
Outcome variables						
Estimated blood loss, mean ± SD (mL)	706.9 ± 674.4	219.2 ± 238.8	0.0001	596.8 ± 492.4	250.0 ± 262.1	0.008
Operative time, mean ± SD (min)	235.9 ± 105.8	265.0 ± 101.5	0.2	256.1 ± 85.4	278.5 ± 104.9	0.4
Postoperative hospital stay, mean ± SD (d)	9.6 ± 6.4	6.8 ± 5.2	0.02	7.7 ± 4.3	6.7 ± 5.8	0.5
Postoperative Clavien–Dindo morbidity ≥ grade 3, *n* (%)	115 (31.6)	2 (7.1)	0.006	4 (18.2)	1 (4.5)	0.2
Conversion rate, *n* (%)	-	1 (3.6)		-	1 (4.5)	
30-days mortality, *n* (%)	2 (0.5)	0 (0)	0.2	0 (0)	0 (0)	
90-day mortality, *n* (%)	4 (1.1)	0 (0)	0.2	0 (0)	0 (0)	
R1 resection, *n* (%)	51 (16.3)	3 (10.7)	0.7	1 (4.5)	3 (13.6)	0.6

**Table 4 cancers-14-01379-t004:** Demographics/confounding variables and clinical outcome variables of patients receiving laparoscopic (LLR) or robotic liver resection (RLR) for colorectal liver metastasis before and after propensity score matching.

Demographics	Before Matching			After Matching		
	LLR *n* = 378	RLR *n* = 28	*p*-Value	LLR *n* = 21	RLR *n* = 21	*p*-Value
Confounding Variables						
Age, mean ± SD (years)	63.8 ± 11.7	61.8 ± 11.0	0.4	62.4 ± 10.6	60.6 ± 10.9	0.6
Gender, *n* (%)			0.06			0.5
Female	161 (42.6)	17 (60.7)		11 (52.4)	13 (61.9)	
Male	217 (57.4)	11 (39.3)		10 (47.6)	8 (38.1)	
ASA score, mean ± SD	2.2 ± 0.6	2.5 ± 0.6	0.001	2.5 ± 0.6	2.5 ± 0.6	0.8
Prior abdominal surgery, *n* (%)	222 (58.7)	16 (57.1)	0.9	15 (71.4)	12 (57.1)	0.3
Liver cirrhosis, *n* (%)	15 (4.2)	3 (10.7)	0.1	5 (23.8)	3 (14.3)	0.7
Prior chemotherapy, *n* (%)	261 (69.4)	19 (70.4)	0.9	15 (71.4)	15 (71.4)	1.0
Number of resected metastases, mean (range)	1.0 (1–3)	1.4 (1–3)	0.003	1.1 (1–3)	1.2 (1–3)	0.7
Diameter of largest metastasis, mean ± SD (cm)	2.4 ± 1.2	2.5 ± 1.1	0.6	2.8 ± 1.3	2.6 ± 1.2	0.6
Extent of liver resection, *n* (%)			0.002			0.7
Major resection (≥3 segments)	177 (47.5)	5 (17.9)		5 (23.8)	3 (14.3)	
Minor resection	196 (52.5)	23 (82.1)		16 (76.2)	18 (85.7)	
Location in deep segments (%)	147 (38.3)	10 (35.7)	0.8	50 (35.2)	6 (27.3)	0.6
Pringle maneuver (%)	106 (28.0)	1 (3.6)	0.003	4 (19)	0 (0)	0.1
Outcome variables						
Estimated blood loss, mean ± SD (mL)	408.6 ± 484.2	219.2 ± 238.8	0.001	777.7 ± 827.1	223.7 ± 255.7	0.04
Operative time, mean ± SD (min)	246.5 ± 115.0	265.0 ± 101.5	0.4	209.7 ± 116.0	271.5 ± 106.3	0.1
Postoperative hospital stay, mean ± SD (d)	5.4 ± 4.7	6.8 ± 5.2	0.1	4.7 ± 3.1	5.1 ± 3.3	0.7
Postoperative Clavien–Dindo morbidity ≥ grade 3, *n* (%)	35 (9.4)	2 (7.1)	1.0	1 (4.8)	0 (0)	1.0
Conversion rate, *n* (%)	33 (8.8)	1 (3.6)	0.5	4 (17.4)	1 (4.3)	0.3
30-day mortality, *n* (%)	0 (0)	0 (0)		0 (0)	0 (0)	
90-day mortality, *n* (%)	4 (1.1)	0 (0)	1.0	0 (0)	0 (0)	
R1 resection, *n* (%)	27 (7.6)	3 (10.7)	0.5	3 (14.3)	3 (14.3,0)	1.0

## Data Availability

Data available upon reasonable request.

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
