# Peer review of "Study: International Multicentric Minimally Invasive Liver Resection for Colorectal Liver Metastases (SIMMILR-CRLM)"

_cancers, 2022, doi:10.3390/cancers14061379_

Round 1

Reviewer 1 Report

In their study, Gumbs et al. present a retrospective analysis of perioperative and long-term outcomes of 1064 liver resections for colorectal liver metastases  performed in 6 international centers depending on open, laparoscopic or robotic approach. Their key finding is, that minimally invasive liver surgery hast short term advantages without compromising oncological results.

As the authors are international experts in minimally invasive liver surgery, this study gives a contemporary overview about the current practice in this field in highly experienced centers. 

General concept comments

The language quality of the study is adequate.

For a retrospective analysis, the methodology extracts the maximum from the presented data by analysing and discussing unmatched and propensity scre matched data of all three groups.

Review:

Nevertheless, some minor questions arise which I would kindly ask the authors to comment on:

Abstract:

The readability of the abstract is low in comparision to the main text, for example “after exculsion criteria” page 1 line 26 is grammatically not correct.

Please mention the total number of cases analysed before matching, as it gives a hint of the overall size of the cohort included.

Page 1 line 30/31: pleas switch to “224 ml vs 778 ml “ as RLR is mentioned before OLR/ LLR.

Page 1 line 33: I doubt, that the term ”follow the Milan criteria” is correct here, as “diameter of largest tumor” is > 25 cmin OLR, LLR an RLR. Please explain the patient selection criteria and reason for exclusion of patients with >= 3 tumors >= 5 cm specifically in methods und use correct terminology throughout the whole article

Introduction:

No comments.

Methods:

Please insert that a retrospective analysis was performed.

Please explain the patients selection criteria and reason for exclusion of patients with >= 3 tumors ³>= 5 cm specifically and use correct terminology throughout the whole article. “Within the Milan crieria“ is not correct in my understanding, as “diameter of largest tumor” is > 25 cmin OLR, LLR an RLR which would be outside Milan criertia.

Please state how written informed consent was obtained exactly for this retrospective analysis, as many patients must have been deceased already at study initiation.

To underline undoubted high experience of minimally invasive/ robotic liver resection in each center, please comment on the proportion of open/ MI/ robotic liver resections each center contributed to the study.

Were repeated liver resections included? If yes, please add their proportion/ group in the results section.

Were ALPPS procedure included?

Were patients after local ablative methods included?

Results:

It would be of high interest, to know how the proportion of open to laparoscopic to robotic liver resection changed over the study period in each center over time. Could data be shown in the results section? This could also underline the discussion/ conclusion paragraph page 14, line 354-359.

 Discussion:

Please discuss that presented data might not be totally representative for the” real world situation” in MI liver surgery, as a selection bias through the contribution of only highly experienced centers is likely.

Specific comments 

Table 1: The decimal places of p-values are not congruent to the other tables, please check.

Table 1-4: Contrary to the methods section, major liver resection is defined as > 2 segments here and not ³ 3 segments. Please check.

Figures 1-6: please consider the use of a continuous marking of the curves per group throughout all figures in order to give the reader a chance to gain an overview more easily ( for example: open group always “triangle”, laparoscopic group always “point”, robotic group always “square”).

Author Response

In their study, Gumbs et al. present a retrospective analysis of perioperative and long-term outcomes of 1064 liver resections for colorectal liver metastases  performed in 6 international centers depending on open, laparoscopic or robotic approach. Their key finding is, that minimally invasive liver surgery hast short term advantages without compromising oncological results.

As the authors are international experts in minimally invasive liver surgery, this study gives a contemporary overview about the current practice in this field in highly experienced centers. 

General concept comments

The language quality of the study is adequate.

For a retrospective analysis, the methodology extracts the maximum from the presented data by analysing and discussing unmatched and propensity scre matched data of all three groups.

Thank you for your thoughtful and constructive comments, we believe the paper has now been significantly improved.

Review:

Nevertheless, some minor questions arise which I would kindly ask the authors to comment on:

Abstract:

The readability of the abstract is low in comparision to the main text, for example “after exculsion criteria” page 1 line 26 is grammatically not correct.

The abstract was refined, it now reads :

Background : This is a retrospective Study: of an International Multicentric cohort after Minimally Invasive Liver Resection (SIMMILR) of Colorectal Liver Metstases (CRLM) from 6 centers.

Methods : Resections were divided by the approach used: Open liver resection (OLR), Laparoscopic liver resection (LLR) and Robotic liver resection (RLR). Patients with macrovascular invasion, ≥  3 tumors measuring more than 3cm or a solitary lesion  >5cm were excluded, and any remaining heterogeneity found was further analyzed after propensity score matching (PSM) to decrease any potential bias.

Results: Prior to matching, 566 patients underwent OLR, 462 LLR and 36 RLR for CRLM. After PSM, 142 patients were in each arm of the OLR vs. LLR group and 22 in the OLR vs. RLR and 21 in the LLR vs. RLR groups. Blood loss, hospital stay, and morbidity rates were all highly statistically significantly increased in the OLR compared to the LLR group, 636mL vs. 353mL, 9 vs. 5 days and 25% vs. 6%, respectively (p<0.001). Only blood loss was significantly decreased when RLR was compared to OLR and LLR, 250mL vs 597mL, and  224 mL vs. 778 mL, p<0.008 and p<0.04, respectively.

Conclusions : SIMMILR indicates that minimally invasive approaches for CRLM that follow the Milan Criteria may have short term advantages, but similar overall and recurrence free survival. Notably, larger studies comparing robotic resections to both OLR and LLR are still needed.

Please mention the total number of cases analysed before matching, as it gives a hint of the overall size of the cohort included.

This was added to the revised abstract :

“Prior to matching, 566 patients were found that underwent OLR, 462 LLR and 36 RLR.”

Page 1 line 30/31: pleas switch to “224 ml vs 778 ml “ as RLR is mentioned before OLR/ LLR.

This error was changed, thank you.

Page 1 line 33: I doubt, that the term ”follow the Milan criteria” is correct here, as “diameter of largest tumor” is > 25 cmin OLR, LLR an RLR. Please explain the patient selection criteria and reason for exclusion of patients with >= 3 tumors >= 5 cm specifically in methods und use correct terminology throughout the whole article.

The definition was clarified in the abstract :

“Patients with macrovascular invasion, ≥  3 tumors measuring more than 3cm or a solitary lesion  >5cm were excluded,”

Introduction:

No comments.

Methods:

Please insert that a retrospective analysis was performed.

This was added.

Please explain the patients selection criteria and reason for exclusion of patients with >= 3 tumors ³>= 5 cm specifically and use correct terminology throughout the whole article. “Within the Milan crieria“ is not correct in my understanding, as “diameter of largest tumor” is > 25 cmin OLR, LLR an RLR which would be outside Milan criertia.

The rationale for using the Milan Criteria and a more complete definition was added to the METHODS:

For the PSM we excluded patients with more than three tumors measuring ≥ 3cm, solitary lesions >5cm and evidence of microvascular invasion. Although originally devised to help in treating patients with hepatocellular cancer, the Milan Criteria have also been used to identify patients with CRLM that would benefit from liver resection, because of this we decided to use this as an exclusion criteria. This was also done to reduce the perceived bias that easier” tumors are removed minimally invasively when compared to open approaches.

This phrase was removed : “We performed propensity score matching (PSM) on only patients with ≤ 3 tumors with all tumors ≤ 5cm to adjust for potential cofounders and, thus, to reduce selection bias.”

This reference was also added : Chiba N, Abe Y, Koganezawa I, Nakagawa M, Yokozuka K, Ozawa Y, Kobayashi T, Sano T, Tomita K, Tsutsui R, Kawachi S. Efficacy of the Milan criteria as a prognostic factor in patients with colorectal liver metastases. Langenbecks Arch Surg. 2021 Jun;406(4):1129-1138.

Please state how written informed consent was obtained exactly for this retrospective analysis, as many patients must have been deceased already at study initiation.

This was added to the METHODS :

On the informed consent that is signed by each patient, it is explained that their anonymous patient data may be used to perform future studies.

To underline undoubted high experience of minimally invasive/ robotic liver resection in each center, please comment on the proportion of open/ MI/ robotic liver resections each center contributed to the study.

The Results section was modified to include this data :

The percentage of cases by center done minimally invasively (laparoscopic and/or robotic) ranged from 13.5 - 87.2% prior to PSM. After matching, the percentage of liver resections done via an open, laparoscopic or robotic approach by center was : 45.4/69.3/0 (center 1), 36.8/3.1/39.6 (center 3), 0/12.9%/0 (center 4), 6.1/1.8/30.2 (center 5) and 11.7/12.9/30.2 (center 6), respectively. No cases from center 2 were included after matching.

Were repeated liver resections included? If yes, please add their proportion/ group in the results section.

This was added to the METHODS :

“Patients who had undergone associating liver partition and portal vein ligation for staged hepatectomy (ALPPS), previous ablation and repeat liver resections were excluded.”

Were ALPPS procedure included?

Please see above.

Were patients after local ablative methods included?

Please see above.

Results:

It would be of high interest, to know how the proportion of open to laparoscopic to robotic liver resection changed over the study period in each center over time. Could data be shown in the results section? This could also underline the discussion/ conclusion paragraph page 14, line 354-359.

As you can now see from this added section :

The percentage of cases by center done minimally invasively (laparoscopic and/or robotic) ranged from 13.5 - 87.2% prior to PSM. After matching, the percentage of liver resections done via an open, laparoscopic or robotic approach by center was : 45.4/69.3/0 (center 1), 36.8/3.1/39.6 (center 3), 0/12.9%/0 (center 4), 6.1/1.8/30.2 (center 5) and 11.7/12.9/30.2 (center 6), respectively. No cases from center 2 were included after matching.

There is quite a bit heterogeneity between centers and not enough robotic cases to make a meaningful comment on this topic when dividing trends by center, as a result, we combined the centers data and created a new Table to show the changing trends of approaches used for liver resections over the years.

Because of this, this was added to the RESULTS :

Figure 1 shows the percentages of approaches (OLR, LLR, RLR) used for the 5 centers included in the PSM over the period of the study (2004-2020).

Figure 1. Changes in percentages of surgical approach used from 2004-2020 of all cases of liver                          resection for colorectal liver metastases (CRLM) after matching. Blue = open liver resection                                 (OLR), Green = laparoscopic liver resection (LLR), Yellow = robotic liver resection (RLR)

 Discussion:

Please discuss that presented data might not be totally representative for the” real world situation” in MI liver surgery, as a selection bias through the contribution of only highly experienced centers is likely.

We actually feel that because of the heterogeneity in our centers, that a real world statement could be presented. Nonetheless, we have added this to the end of the DISCUSSION:

“Although there is significant heterogeneity between the approaches used by center, the centers included in this study are referral centers and are highly experienced in minimally invasive surgery, as a result, the presented data might not be totally representative for real world situations.”

Specific comments 

Table 1: The decimal places of p-values are not congruent to the other tables, please check.

This was changed to match the other Tables, thank you.

Table 1-4: Contrary to the methods section, major liver resection is defined as > 2 segments here and not ³ 3 segments. Please check.

This was changed on the Tables to read “≥ 3”, thank you!

Figures 1-6: please consider the use of a continuous marking of the curves per group throughout all figures in order to give the reader a chance to gain an overview more easily ( for example: open group always “triangle”, laparoscopic group always “point”, robotic group always “square”).

This was changed in all 6 Graphs, thank you.

Reviewer 2 Report

This is a multicenter retrospective study comparing short and long-term outcomes of patients undergoing open, laparoscopic, and robotic liver resections for colorectal cancer liver metastases in number ≤3 and size ≤ 5 cm, using a propensity score match analysis.

In my opinion the study is novel, and the topic is interesting, although it needs major revisions in the current version. The hypothesis must be clearly stated, and the methodology report improved.

Abstract

  • Please state that this is a retrospective study (both in the abstract and in the summary).
  • Use the standard structure for the abstract (background, methods, results, conclusions) to make it more readable.
  • In the abstract it is stated that patients with ≥3 tumors measuring ≥5cm were excluded: this is not what is stated in the summary and in the main text, as patients with ≤3 metastases and ≤5 cm were included. Please be precise on this, this is very important. Specify that the number and the size are those of the metastases (why tumors?) [major point]
  • In the results: “Blood loss, hospital stay, and morbidity rates were all highly statistically significantly increased in the OLR group, 636mL vs. 353mL, 9 vs. 5 days and 25% vs. 6%, respectively (p<0.001)” it is not clear. Which one was the group OLR was compared with?
  • “Only blood loss was significantly decreased when RLR was compared to OLR and LLR, 250mL vs 597mL, and 778 mL 224 mL, p<0.008 and p<0.04, respectively.” It seems that RLR has more EBL than LLR, please put everything in the correct order inverting “778 mL vs. 224 mL” (respectively).

Materials and methods

  • The requirements for participating in the study do not seem to be very clear. Each center had a team of 6 surgeons (2 pioneers, 2 early adopters and 2 FT)? Was it true for all the period 2004-2020 for all the 6 centers? Only 3 surgeons in each team performed also robotic resections? Please clarify.
  • Please state if the study followed any retrospective study reporting guidelines (e.g. STROBE).
  • “For this study we excluded patients with more than three tumors measuring ≥ 5cm”. This is in contrast with what is stated in the PSM paragraph: “only patient’s with ≤ 3 tumors with all tumors ≤ 5cm to adjust for potential cofounders”. It is not clear if patients with metastases measuring 5 cm are included or excluded. Then, why the term “tumor” and not “metastases”? Furthermore, is the 5 cm a cut-off value resulting from the total size of the metastases? It is also not clear if both the criteria must be satisfied at the same time (eg. ≤ 3 metastases whose total size is ≤5 cm). In my opinion this is a very important point. The inclusion criteria must be stated very clearly and precisely [major point].
  • How did you measure the number and size of the metastases? Are the number given in the tables based on a preoperative or pathological exam? The doubt rises because it is written: “number of resected tumors, maximum size of resected lesion” in the PSM paragraph. [major point]
  • Because the overall survival and recurrence-free survival are primary endpoints, further definitions should be given. Did all the patients in the study undergo a liver resection with a curative intent? When did you started to count the OS and RFS periods? This is also a very crucial point since comparisons of survival outcomes are provided [major point].
  • Which statistical test was used to compare the Kaplan-Meier curves?

Results

  • Table 1: While there is one p-value for the open and laparoscopic comparison, there is only one p-value for the Robot group, why? It is not clear which group the robot one is compared with (the p-values should be three for pairwise comparisons between three groups). Furthermore, I would state and clarify in the caption that this population is before PSM and before the inclusion/exclusion criteria were applied (number/size).
  • Survival data: I found a bit hard to read different year-OS and RFS for different comparisons (e.g. 1- 2- 3- 5-year OS/RFS for a comparison, 1-, 2- 3- for another comparison and 1- 3- 5- for another one). I would always keep the same pattern for clarity.
  • All the survival figures should have the number of the total patients at the beginning of each interval, stated under the horizontal axis, in order to easily show the number of events happened at the end of each interval (12 months, 24 months…).
  • Why there is “5-year”, “3-year” in the survival curves captions? The survival curves are drawn in a continuous fashion.
  • Figure 1: not readable because it is too much zoomed out (busy). It is drawn on a 12.5 year period (150 months/12)! The single lines are not distinguishable. Please draw it on a shorter period in order to make the lines be readable (e.g. on a 5 year period).

Discussion

  • “LLR seems to have benefits in improved lengths of hospitalization and decreased complication rates…” I would add “compared with the OLR”.
  • “Although RFS tended to be significantly decreased after RLR when compared to either OLR or LLR”: please specify that it was not statistically significant.
  • “Complications occurred in 4-6% of the laparoscopic patients, 0-22% in the robotic cohort and 25-35% in the open cohort”: what are you referring to? All the complications? The major complications (CD≥3)?
  • “The range of the traditional learning curve as described by Brown and Geller…by way of hand-held instruments.”: I cannot see the relevance of this in this study article.
  • Please, discuss the limits of this study. [major point]

Conclusions

  • “LLR and RLR may be able to have similar major resection rates as OLR.”: this is not proven by this study. Please erase or amend according to the findings of the study. [major point]
  • “Longer term follow-up is still needed in the robotic arm”: and higher numbers of patients.
  • “The results are so compelling…as more MI HPB FT surgeons are trained.”: this seems more an opinion of the authors than a conclusion based on the study results. Please erase or amend according to the results. [major point]
  • “Minimally invasive liver resection may enable equivalent parenchymal sparing hepatectomies to OLR in tumors that follow the Milan Criteria.”: I am concerned this sentence is also not completely based on the results of the study (in the abstract as well). [major point]

Author Response

This is a multicenter retrospective study comparing short and long-term outcomes of patients undergoing open, laparoscopic, and robotic liver resections for colorectal cancer liver metastases in number ≤3 and size ≤ 5 cm, using a propensity score match analysis.

In my opinion the study is novel, and the topic is interesting, although it needs major revisions in the current version. The hypothesis must be clearly stated, and the methodology report improved.

Thank you for your thoughtful and meticulous comments, we believe the paper has now been profoundly improved.

Abstract

  • Please state that this is a retrospective study (both in the abstract and in the summary).

This was added

  • Use the standard structure for the abstract (background, methods, results, conclusions) to make it more readable.

This was added, please see the new edited ABSTRACT

Background : This is a retrospective Study: of an International Multicentric cohort after Minimally Invasive Liver Resection (SIMMILR) of Colorectal Liver Metastases (CRLM) from 6 centers.

Methods : Resections were divided by the approach used: Open liver resection (OLR), Laparoscopic liver resection (LLR) and Robotic liver resection (RLR). Patients with macrovascular invasion, > 3 tumors measuring more than 3cm or a solitary lesion  >5cm were excluded, and any remaining heterogeneity found was further analyzed after propensity score matching (PSM) to decrease any potential bias.

Results: Prior to matching, 566 patients underwent OLR, 462 LLR and 36 RLR for CRLM. After PSM, 142 patients were in each arm of the OLR vs. LLR group and 22 in the OLR vs. RLR and 21 in the LLR vs. RLR groups. Blood loss, hospital stay, and morbidity rates were all highly statistically significantly increased in the OLR compared to the LLR group, 636mL vs. 353mL, 9 vs. 5 days and 25% vs. 6%, respectively (p<0.001). Only blood loss was significantly decreased when RLR was compared to OLR and LLR, 250mL vs 597mL, and  224 mL vs. 778 mL, p<0.008 and p<0.04, respectively.

Conclusions : SIMMILR indicates that minimally invasive approaches for CRLM that follow the Milan Criteria may have short term advantages, but similar overall and recurrence free survival. Notably, larger studies comparing robotic resections to both OLR and LLR are still needed.

  • In the abstract it is stated that patients with ≥3 tumors measuring ≥5cm were excluded: this is not what is stated in the summary and in the main text, as patients with ≤3 metastases and ≤5 cm were included. Please be precise on this, this is very important. Specify that the number and the size are those of the metastases (why tumors?) [major point]

This was corrected in the ABSTRACT and BODY of the manuscript. Thank you for shining a light on this error.

  • In the results: “Blood loss, hospital stay, and morbidity rates were all highly statistically significantly increased in the OLR group, 636mL vs. 353mL, 9 vs. 5 days and 25% vs. 6%, respectively (p<0.001)” it is not clear. Which one was the group OLR was compared with?

It was compared with the laparoscopic group. Thank you for picking this up! This was added to the abstract :

“… compared to the LLR group,….”

  • “Only blood loss was significantly decreased when RLR was compared to OLR and LLR, 250mL vs 597mL, and 778 mL 224 mL, p<0.008 and p<0.04, respectively.” It seems that RLR has more EBL than LLR, please put everything in the correct order inverting “778 mL vs. 224 mL” (respectively).

This error was corrected, please see above and again, THANK YOU.

Materials and methods

  • The requirements for participating in the study do not seem to be very clear. Each center had a team of 6 surgeons (2 pioneers, 2 early adopters and 2 FT)? Was it true for all the period 2004-2020 for all the 6 centers? Only 3 surgeons in each team performed also robotic resections? Please clarify.

We see the confusion. This section was modified to clarify this.

All hepatectomies for colorectal liver metastases done by  single surgeon with experience in both open and minimally invasive liver resection at 6 different international centers (2 USA, 1 Italy, 1 France, 1 Germany and 1 Taiwan) were divided into open, laparoscopic and robotic cohorts and compared. Minimum case requirements for participating HPB surgeons in this study was the completion of at least, 50 laparoscopic and/or robotic hepatectomies and, at least, 40 hepatectomies for CRLM.

  • Please state if the study followed any retrospective study reporting guidelines (e.g. STROBE).

This was added to the end of the METHODS :

“The STROBE statement checklist for reporting of observational studies was used during the drafting and editing of the manuscript.”

  • “For this study we excluded patients with more than three tumors measuring ≥ 5cm”. This is in contrast with what is stated in the PSM paragraph: “only patient’s with ≤ 3 tumors with all tumors ≤ 5cm to adjust for potential cofounders”. It is not clear if patients with metastases measuring 5 cm are included or excluded. Then, why the term “tumor” and not “metastases”? Furthermore, is the 5 cm a cut-off value resulting from the total size of the metastases? It is also not clear if both the criteria must be satisfied at the same time (eg. ≤ 3 metastases whose total size is ≤5 cm). In my opinion this is a very important point. The inclusion criteria must be stated very clearly and precisely [major point].

When possible, the term tumor was switched for metastasis throughout the abstract, manuscript and tables.

Our previous definition for the Milan criteria was poor and it has been modified throughout the manuscript. As seen above in the ABSTRACT, but also in the body of the manuscript. Additionally are full inclusion criteria was enhanced in the METHODS section :

For the PSM we excluded patients with more than three metastases measuring ≥ 3cm, solitary lesions >5cm and evidence of microvascular invasion. Although originally devised to help in treating patients with hepatocellular cancer, the Milan Criteria have also been used to identify patients with CRLM that would benefit from liver resection, because of this we decided to use this as an exclusion criteria [39]. This was also done to reduce the perceived bias that easier” tumors are removed minimally invasively when compared to open approaches. Patients who had undergone associating liver partition and portal vein ligation for staged hepatectomy (ALLPS), previous ablation and repeat liver resections were excluded.

  • How did you measure the number and size of the metastases? Are the number given in the tables based on a preoperative or pathological exam? The doubt rises because it is written: “number of resected tumors, maximum size of resected lesion” in the PSM paragraph. [major point]

“Metastasis size was based on final pathological result.” As a result, this statement was added to the Propensity Score Matching section of METHODS.

The term lesion was changed for metastasis throughout the Abstract, body of the manuscript and Tables when additional identifying information such as “secondary” was not present.

  • Because the overall survival and recurrence-free survival are primary endpoints, further definitions should be given. Did all the patients in the study undergo a liver resection with a curative intent? When did you started to count the OS and RFS periods? This is also a very crucial point since comparisons of survival outcomes are provided [major point].

This was clarified in the Data Analysis section :

“OS was calculated from the date of liver resection. RFS was calculated from whichever date of diagnosis of recurrence was earliest. Recurrence was either diagnosed from rising serum tumor markers, radiographic examination, or upon positive histological confirmation from percutaneous or surgical biopsy.”

  • Which statistical test was used to compare the Kaplan-Meier curves?

The Log Rank test was used. This was added to the Data Analysis section :

“Prism 8: GraphPad software (https://www.graphpad.com/scientific-software/prism/) was used to generate Kaplan-Meier curves, and the Log-rank (Mantel Cox) test was used to calculate p-values.”

Results

  • Table 1: While there is one p-value for the open and laparoscopic comparison, there is only one p-value for the Robot group, why? It is not clear which group the robot one is compared with (the p-values should be three for pairwise comparisons between three groups). Furthermore, I would state and clarify in the caption that this population is before PSM and before the inclusion/exclusion criteria were applied (number/size).

The caption was modified and now reads :

Table 1. Demographics/confounding variables of patients receiving open (OLR), laparoscopic liver resection (LLR) or robotic liver resection (RLR) for colorectal liver metastasis before propensity score matching and before the exclusion criteria were applied.

Another column was added for the comparison between the Open and Robotic arms. Initially only Open vs. Laparoscopic and Laparoscopic vs. Robotic data was presented. Additionally information on location in the Deep segments and utilization of the Pringle maneuver was added to all Tables.

To further clarify this modification, the results section now reads :

         When unmatched patients undergoing OLR were compared to LLR, there were no significant differences between age, sex distribution, BMI and carcinoembryonic antigen (CEA) levels. Although, increases in ASA scores were statistically significant (p=0.03) and an increased history of previous abdominal surgery, history of neoadjuvant chemotherapy, mean metastasis size and number of metastases was highly statistically significant. The rate of cirrhosis was highly significantly more prevalent in the LLR group compared to the OLR group. Lastly, lesions were significantly more prevalent in the deeper segments in the OLR group, p= 0.03. No differences were noted in the utilization of the Pringle maneuver.

         When unmatched patients who had undergone OLR were compared to RLR, ASA class and presence of cirrhosis was significantly higher in the RLR group, p= 0.02 and 0.03, respectively), however, significantly more patients received neoadjuvant chemotherapy, had undergone previous surgery and had on average higher pre-oerative CEA serum levels, p= 0.03, 0.04 and 0.08, respectively. Furthermore more metastases that were larger were removed in the OLR group, p= 0.003 and 0.007, respectively. Although lesions tended to be in the deeper segments more often in the OLR group, this did not attain statistical significance, p=0.06. The Pringle maneuver was used significantly more frequently in the OLR group when compared to the RLR group, p= 0.004)

         When unmatched patients who had undergone LLR were compared to RLR, the ASA class was statistically higher in the RLR group, p= 0.001, and the Pringle maneuver was used statistically more often in the LLR group, p=0.02.

  • Survival data: I found a bit hard to read different year-OS and RFS for different comparisons (e.g. 1- 2- 3- 5-year OS/RFS for a comparison, 1-, 2- 3- for another comparison and 1- 3- 5- for another one). I would always keep the same pattern for clarity.

The data was changed to 1, 3 and 5- year reporting or to 1 and 3- year reporting  to help standardize the article. This section now reads :

3.4. Long-term survival data

After PSM overall survival (OS) at 1, 3 and 5-years was 80.2%, 53.0% and 44.1% with a median survival of 37 months after OLR, compared to 91.4%, 64.6%% and 42.6% with a median survival of 48 months after LLR, respectively (p=0.401) (Figure 2). Only OS at 1 and 3-years could be calculated when OLR and RLR were compared after PSM and was 100% and 27 months after OLR, compared to 93.8% and 32.1% with a median survival of 34 months after RLR, respectively (p=0.581) (Figure 3). Similarly, OS at 1 and 3-years was 100% and 73.8% and with a median survival of 61 months after LLR, compared to 94.1% and 36.7% after RLR with a median survival of 34 months, respectively (p=0.201) (Figure 4).

         Recurrence free survival (RFS) at 1, 3 and 5-years was 75.2%, 58.1% and 24.1%  with a median survival of 45 months after OLR, compared to 90.9%, 47.3% and 17.8% with a median survival of 35 months after LLR, respectively (p=0.710) (Figure 5). RFS at 1 and 3 years was 88.9% and 47.6% with a median follow-up of 23 months after OLR compared to 52.7% and 10.5% with a median follow-up of 14 moths after RLR, respectively (p=0.062) (Figure 6). RFS at 1 and 3-years was 60.0% and 36.0% after LLR with a median follow-up of 22 months, compared to 52.7% and 9.4% with a median follow-up of 13 months after RLR, respectively (p=0.079) (Figure 7).

  • All the survival figures should have the number of the total patients at the beginning of each interval, stated under the horizontal axis, in order to easily show the number of events happened at the end of each interval (12 months, 24 months…).

All survival curves have been modified to show the total number of patients at 12 month intervals depending on the curve. The captions now have these phrases added to the end of them to highlight this,

Figure 2 : Total number of patients at 12, 24, 36, 48 and 60 months noted at the bottom of the graph according to surgical approach used.

Figure 3 : Total number of patients at 12, 24, 36 and 48 months noted at the bottom of the graph according to surgical approach used.

Figure 4 : Total number of patients at 12, 24, 36, 48 and 60 months noted at the bottom of the graph according to surgical approach used.

Figure 5 : Total number of patients at 12, 24, 36, 48 and 60 months noted at the bottom of the graph according to surgical approach used.

Figure 6 : Total number of patients at 12, 24, 36 and 48 months noted at the bottom of the graph according to surgical approach used.

Figure 7 : Total number of patients at 12, 24, 36 and 48 months noted at the bottom of the graph according to surgical approach used.

  • Why there is “5-year”, “3-year” in the survival curves captions? The survival curves are drawn in a continuous fashion.

The number and year was removed from the captions.

  • Figure 1: not readable because it is too much zoomed out (busy). It is drawn on a 12.5 year period (150 months/12)! The single lines are not distinguishable. Please draw it on a shorter period in order to make the lines be readable (e.g. on a 5 year period).

Figure 1, which is now Figure 2 was changed to only show data on a 5 year period.

Discussion

  • “LLR seems to have benefits in improved lengths of hospitalization and decreased complication rates…” I would add “compared with the OLR”.

This was modified, the sentence now reads :

“LLR seems to have benefits in improved lengths of hospitalization and decreased complication rates, when compared with the OLR.”

  • “Although RFS tended to be significantly decreased after RLR when compared to either OLR or LLR”: please specify that it was not statistically significant.

This was changed and now reads :

“Although RFS tended to be decreased after RLR when compared to either OLR or LLR this was not statistically significant (p=0.06 and 0.08, respectively). Additionally, the small numbers in each arm and fact that OS is not significantly different make it unlikely that this difference would be found in larger series.”

  • “Complications occurred in 4-6% of the laparoscopic patients, 0-22% in the robotic cohort and 25-35% in the open cohort”: what are you referring to? All the complications? The major complications (CD≥3)?

Yes, after major complications, this was clarified and the sentence now reads :

"Clavien-Dindo grade 3 complications occurred in 4-6% of the laparoscopic patients, 0-22% in the robotic cohort and 25-35% in the open cohort and although this difference was statistically significant only when the LLR was compared to the OLR group, these rates are well within the range published in the literature of 5-55% after laparoscopic hepatectomy, 17-20% after robotic resection and 17-61% after open hepatectomy.”

  • “The range of the traditional learning curve as described by Brown and Geller…by way of hand-held instruments.”: I cannot see the relevance of this in this study article.

This is relevant because of the heterogeneity between case types in the centers included in the study, we wanted to explain the inclusion criteria of the centers.

The sentence has been modified to highlight this point:

“All participants had to have done, at least 50 minimally invasive hepatectomies to be included in this trial because the range of the traditional learning curve has been described by Brown and Geller to range between 45 and 60 [31, 58].”

  • Please, discuss the limits of this study. [major point]

This was added at the end of the DISCUSSION to address this :

         “Although there is significant heterogeneity between the approaches used by center, the centers included in this study are referral centers and are highly experienced in minimally invasive surgery, as a result, the presented data might not be totally representative for real world situations. Another limitation of the study is the lack of longer follow-up and low numbers, particularly, in the robotic group. Additionally, the case number of RLR is small and the cases are from only 3 centers. This is due to the fact that robotic surgery of the liver is still in its infancy. We had strict criteria inclusion and exclusion criteria for our PSM and needed to include non-robotic centers to get enough patients to make any meaningful comments on the various approaches to liver surgery. Before matching, Pringle maneuver was used significantly more during OLR and LLR when compared to RLR, after matching this difference was no longer statistically significant. The observation that patients had on average 540mL more blood loss after LLR and RLR after matching is probably due to the fact that more patients had tended to have cirrhosis in the LLR arm.

         In the future, as RLR becomes more common we believe that it will indeed be possible to do a better multicentric study where each center has enough open, laparoscopic and robotic liver resection, however, we are not there yet. Importantly, we believe that future liver surgeons should become versed in all 3 approaches and that future studies on liver surgery should include data from all 3 techniques and not solely be limited to binary analyses (i.e. OLR vs RLR).

Conclusions

  • “LLR and RLR may be able to have similar major resection rates as OLR.”: this is not proven by this study. Please erase or amend according to the findings of the study. [major point]

This sentence was removed.

  • “Longer term follow-up is still needed in the robotic arm”: and higher numbers of patients.

This correct point was added and the sentence now reads :

"Regardless of whether open, laparoscopic or robotic hepatectomy is performed, no significant differences in OS or RFS are found and survival rates are similar, however, more patients and longer term follow-up are still needed in the robotic arm.”

  • “The results are so compelling…as more MI HPB FT surgeons are trained.”: this seems more an opinion of the authors than a conclusion based on the study results. Please erase or amend according to the results. [major point]

This sentence was removed.

  • “Minimally invasive liver resection may enable equivalent parenchymal sparing hepatectomies to OLR in tumors that follow the Milan Criteria.”: I am concerned this sentence is also not completely based on the results of the study (in the abstract as well). [major point]

We hope that with the clarification of the definition of the Milan Criteria used in the ABSTRACT and Manuscript, and improvements made throughout the manuscript regarding exclusion and inclusion criteria and above clarifications that this sentence is now more reasonable.

The Final CONCLUSION now reads :

         “The Study: of International Multicentric Minimally Invasive Liver Resection  (SIMMILR) is a retrospective PSM study of CRLM, which reveals that LLR may result in decreased length of hospital stay and complication rates in the compared to OLR. Both minimally invasive approaches (laparoscopic and robotic) seem to have the added benefit of significantly decreased blood loss when compared to OLR without significantly increasing operative times. Regardless of whether open, laparoscopic or robotic hepatectomy is performed, no significant differences in OS or RFS are found and survival rates are similar, however, more patients and longer term follow-up are still needed in the robotic group. Minimally invasive liver resection may enable equivalent parenchymal sparing hepatectomies to OLR in metastases that follow the Milan Criteria.”

Reviewer 3 Report

This study conducted retrospective multi-centric international research that compared open, laparoscopic and robotic liver resection for colorectal liver metastases matching with Milan criteria. The authors concluded that minimally invasive liver resection may have short term advantages, but no differences in overall and recurrence-free survival. This study may be the first that examined the outcomes after three different liver resections, but have some drawbacks.

  1. In propensity score matching (PSM), the prevalence of previous abdominal surgery was chosen as a confounding variable. The definition of the previous abdominal surgery is unclear. Did the previous abdominal surgery contain all surgery regarding abdomen such colorectal resection, hepatectomy, et al? There is a difference in the impact on the short and long-term outcomes.
  2. The tumor location has also the great impact on the difficulty of liver resection, especially segments 7 and 8. The authors should add the factor to the predictors in PSM.

Author Response

This study conducted retrospective multi-centric international research that compared open, laparoscopic and robotic liver resection for colorectal liver metastases matching with Milan criteria. The authors concluded that minimally invasive liver resection may have short term advantages, but no differences in overall and recurrence-free survival. This study may be the first that examined the outcomes after three different liver resections, but have some drawbacks.

 Thank you for your thoughtful and constructive comments.

  1. In propensity score matching (PSM), the prevalence of previous abdominal surgery was chosen as a confounding variable. The definition of the previous abdominal surgery is unclear. Did the previous abdominal surgery contain all surgery regarding abdomen such colorectal resection, hepatectomy, et al? There is a difference in the impact on the short and long-term outcomes.

Patients who had previous liver resections were excluded from the PSM. This section was added to the METHODS section to highlight this.

“For the PSM we excluded patients with more than three metastases measuring ≥ 3cm, solitary metastasiss >5cm and evidence of microvascular invasion. Although originally devised to help in treating patients with hepatocellular cancer, the Milan Criteria have also been used to identify patients with CRLM that would benefit from liver resection, because of this we decided to use this as an exclusion criteria [39]. This was also done to reduce the perceived bias that easier” tumors are removed minimally invasively when compared to open approaches. Patients who had undergone associating liver partition and portal vein ligation for staged hepatectomy (ALLPS), previous ablation and repeat liver resections were excluded.”

  1. The tumor location has also the great impact on the difficulty of liver resection, especially segments 7 and 8. The authors should add the factor to the predictors in PSM.

The definition used for Deep segments was added to the METHODS :

“Lesions in the Deep or Deeper segments were defined as metastases in segments 4B, 7or 8.”

The information on the Deep segments was added to the Tables and this section was added to the RESULTS to address this :

When unmatched patients undergoing OLR were compared to LLR, there were no significant differences between age, sex distribution, BMI and carcinoembryonic antigen (CEA) levels. Although, increases in ASA scores were statistically significant (p=0.03) and an increased history of previous abdominal surgery, history of neoadjuvant chemotherapy, mean metastasis size and number of metastases was highly statistically significant. The rate of cirrhosis was highly significantly more prevalent in the LLR group compared to the OLR group. Lastly, lesions were significantly more prevalent in the deeper segments in the OLR group, p= 0.03. No differences were noted in the utilization of the Pringle maneuver.

         When unmatched patients who had undergone OLR were compared to RLR, ASA class and presence of cirrhosis was significantly higher in the RLR group, p= 0.02 and 0.03, respectively), however, significantly more patients received neoadjuvant chemotherapy, had undergone previous surgery and had on average higher pre-oerative CEA serum levels, p= 0.03, 0.04 and 0.08, respectively. Furthermore more metastases that were larger were removed in the OLR group, p= 0.003 and 0.007, respectively. Although lesions tended to be in the deeper segments more often in the OLR group, this did not attain statistical significance, p=0.06. The Pringle maneuver was used significantly more frequently in the OLR group when compared to the RLR group, p= 0.004)

         When unmatched patients who had undergone LLR were compared to RLR, the ASA class was statistically higher in the RLR group, p= 0.001, and the Pringle maneuver was used statistically more often in the LLR group, p=0.02.

3.2. Differences between open (OLR) and robotic liver resection (RLR)

         Table 3 describes and compares demographics, confounding variables and clinical outcome variables of patients undergoing OLR and RLR for colorectal liver metastases that follow the Milan Criteria before and after propensity score matching (PSM). During PSM 22 patients were matched 1:1 in each group. After PSM no differences were noted in average metastasis size, number of resected metastatic lesions, percentages of major hepatectomies, location in the Deep segments or utilization of the Pringle maneuver between the 2 approaches. Although before matching patients in the RLR group had a significantly higher prevalence of liver cirrhosis (10.7 % vs. 0.3 %; p<0.001) and severe co-morbidities indicated by significantly higher mean ASA score (2.5 vs. 2.1; p<0.001); and metastases were found significantly more often in the Deep segments (44.0% vs. 35.7%; p=0.04) and the Pringle maneuver was used more frequently in the OLR group (25% vs. 3.6%; p=0.009), these differences were insignificant after the matching process. Notably, after PSM estimated blood loss remained significantly lower in cases of RLR when compared to OLR (596.8 mL vs. 250.0 mL; p<0.008).

3.3. Differences between laparoscopic (LLR) and robotic liver resection (RLR)

         Table 4 describes demographics and clinical outcome variables of LLR and RLR. Although before matching patients in the RLR group had a significantly higher number of metastases resected (1.4 vs.1.0; p=0.003); and significantly more major resections (47.5% vs. 17.9%; p=0.002) and the Pringle maneuver  (28.0% vs. 3.6%; p=0.003)was used more frequently in the LLR group, these differences were insignificant after the matching process. During PSM 21 patients were matched 1:1 in each group. Notably, estimated blood loss was significantly less in RLR (219.2 ml vs. LLR group 408.6 ml; p<0.001) before PSM and remained less in the RLR group after PSM (RLR 223.7 ml vs. LLR 777.7 ml; p=0.04).

This was added to the DISCUSSION :

         Although there is significant heterogeneity between the approaches used by center, the centers included in this study are referral centers and are highly experienced in minimally invasive surgery, as a result, the presented data might not be totally representative for real world situations. Another limitation of the study is the lack of longer follow-up and low numbers, particularly, in the robotic group. Additionally, the case number of RLR is small and the cases are from only 3 centers. This is due to the fact that robotic surgery of the liver is still in its infancy. We had strict criteria inclusion and exclusion criteria for our PSM and needed to include non-robotic centers to get enough patients to make any meaningful comments on the various approaches to liver surgery. Before matching, Pringle maneuver was used significantly more during OLR and LLR when compared to RLR, after matching this difference was no longer statistically significant. The observation that patients had on average 540mL more blood loss after LLR and RLR after matching is probably due to the fact that more patients had tended to have cirrhosis in the LLR arm.

         In the future, as RLR becomes more common we believe that it will indeed be possible to do a better multicentric study where each center has enough open, laparoscopic and robotic liver resection, however, we are not there yet. Importantly, we believe that future liver surgeons should become versed in all 3 approaches and that future studies on liver surgery should include data from all 3 techniques and not solely be limited to binary analyses (i.e. OLR vs RLR).

The CONCLUSION was also been heavily edited, with some of the more ambitious statements removed :

         The Study: of International Multicentric Minimally Invasive Liver Resection  (SIMMILR) is a retrospective PSM study of CRLM, which reveals that LLR may result in decreased length of hospital stay and complication rates in the compared to OLR. Both minimally invasive approaches (laparoscopic and robotic) seem to have the added benefit of significantly decreased blood loss when compared to OLR without significantly increasing operative times. Regardless of whether open, laparoscopic or robotic hepatectomy is performed, no significant differences in OS or RFS are found and survival rates are similar, however, more patients and longer term follow-up are still needed in the robotic group. Minimally invasive liver resection may enable equivalent parenchymal sparing hepatectomies to OLR in metastases that follow the Milan Criteria.

Reviewer 4 Report

Thank you for your submitting this interesting manuscript.

Open and MIS (mainly LLR) comparison is reasonable and the results are understandable. However, the results of comparisons to RLR of OLR and LLR are controversial and should be presented in more deliberate way, and with less emphasis just as supporting data. 

The case number of RLR is small and the cases are from only 3 centers. Also, the follow-up period of them are short, which means thay are recent new cases. I'm afraid that the comparisons of these cases to OLR and LRL cases from all 6 centers during long period might be inappropriate, even after PSM. The authors can compare the RLR cases to OLR or LLR cases from the  same 3 centers during same peroid with PSM. Also, the blood loss of LRL after matching in the comparison to RLR are almost 800ml, which seems too large amount of bleeding for usual LLR. (At least, the points above listed should be discussed. )

Author Response

Thank you for your submitting this interesting manuscript.

Open and MIS (mainly LLR) comparison is reasonable and the results are understandable. However, the results of comparisons to RLR of OLR and LLR are controversial and should be presented in more deliberate way, and with less emphasis just as supporting data. 

The case number of RLR is small and the cases are from only 3 centers. Also, the follow-up period of them are short, which means thay are recent new cases. I'm afraid that the comparisons of these cases to OLR and LRL cases from all 6 centers during long period might be inappropriate, even after PSM. The authors can compare the RLR cases to OLR or LLR cases from the  same 3 centers during same peroid with PSM. Also, the blood loss of LRL after matching in the comparison to RLR are almost 800ml, which seems too large amount of bleeding for usual LLR. (At least, the points above listed should be discussed. )

DISCUSSION

We understand your concerns and attempted to address them by adding the use of Pringle maneuver to the Tables, RESULTS and DISCUSSION and better highlighting the limitations of the study at the end of the DISCUSSION section.

         “Although there is significant heterogeneity between the approaches used by center, the centers included in this study are referral centers and are highly experienced in minimally invasive surgery, as a result, the presented data might not be totally representative for real world situations. Another limitation of the study is the lack of longer follow-up and low numbers, particularly, in the robotic arm. Additionally, the case number of RLR is small and the cases are from only 3 centers. This is due to the fact that robotic surgery of the liver is still in its infancy. We had strict criteria inclusion and exclusion criteria for our PSM and needed to include non-robotic centers to get enough patients to make any meaningful comments on the various approaches to liver surgery. Before matching, Pringle maneuver was used significantly more during OLR and LLR when compared to RLR, after matching this difference was no longer statistically significant. The observation that patients had on average 540mL more blood loss after LLR and RLR after matching is probably due to the fact that more patients had tended to have cirrhosis in the LLR arm.

         In the future, as RLR becomes more common we believe that it will indeed be possible to do a better multicentric study where each center has enough open, laparoscopic and robotic liver resection, however, we are not there yet. Importantly, we believe that future liver surgeons should become versed in all 3 approaches and that future studies on liver surgery should include data from all 3 techniques and not solely be limited to binary analyses (i.e. OLR vs RLR).

CONCLSUIONS

Additionally, these phrases were removed from the CONCLUSIONS :

“LLR and RLR may be able to have similar major resection rates as OLR."

“The results are so compelling that it seems possible that international multi-centered randomized-controlled trials comparing open and MIS hepatectomies wont be feasible as increasing numbers of patients demand minimally invasive liver surgery and as more MI HPB FT surgeons are trained.”

And this phrase added :

“Regardless of whether open, laparoscopic or robotic hepatectomy is performed, no significant differences in OS or RFS are found and survival rates are similar, however, more patients and longer term follow-up are still needed in the robotic arm.”

In summary, the CONCLUSION now reads :

         “The Study: of International Multicentric Minimally Invasive Liver Resection  (SIMMILR) is a retrospective PSM study of CRLM, which reveals that LLR may result in decreased length of hospital stay and complication rates in the compared to OLR. Both minimally invasive approaches (laparoscopic and robotic) seem to have the added benefit of significantly decreased blood loss when compared to OLR without significantly increasing operative times. Regardless of whether open, laparoscopic or robotic hepatectomy is performed, no significant differences in OS or RFS are found and survival rates are similar, however, more patients and longer term follow-up are still needed in the robotic arm. Minimally invasive liver resection may enable equivalent parenchymal sparing hepatectomies to OLR in metastases that follow the Milan Criteria.”

Reviewer 5 Report

I have read with interest this manuscript that concerns the results of a multicentric retrospective study on open versus minimally invasive surgery for CRLM. While the topic might be of general interest to the readers of Cancers, I believe that there are some flaws that strongly limit the quality of this paper.

  1. The manuscript should be reviewed for proofreading and English grammar. There are also several typos throughout the paper and tables.
  2. The title should be modified because it is unusual to start with “Study: … “. Similarly, the first sentence of the abstract should be modified.
  3. In the abstract and in the main text the authors refer to the Milan Criteria as one of the inclusion criteria for this study. First, the authors should be better detail the rationale of this choice, meaning the exclusion of patients with more than 3 tumors measuring more than 5 cm. Second, such definition refers to transplantation criteria for HCC. This definition and concept should not be used for CRLM. Third, it is unclear the real application of this definition in this study since later in the text the authors report a case of surgery for a 17-cm single lesion, which should be excluded according with their CLM Milan Criteria.
  4. The introduction and the discussion include some paragraphs with specific citations that show a kind of enthusiastic way of writing that anticipates the results of the study. The paper should be written with more balance.
  5. References: REF number 50 is missing; more importantly, there are approximately 20 references (31%) from the same authors. Self-citation should be limited to 15-20%. Besides, some of these citations are case reports, that might be omitted.
  6. Most of the introduction concerns the history of published consensus conferences on MIS in liver surgery, which I believe is not pertinent and should be, if important to the authors, moved to the discussion.
  7. In particular, the paragraph that goes from row 55 to row 62 is critical. I believe that the surgical approach should not change the indications in liver surgery. Never. The authors should change the word “indications” with something else. Otherwise, the risk is that – as already done in the past – MIS is used to remove asymptomatic small benign liver tumors that should not be touched (nor previously and for sure nor in 2022).
  8. Along the point of benign tumors, I suggest to the authors to change the sentence at rows 315-317. It is unacceptable to justify surgery for benign liver tumors for training reasons. The training in surgery is not a race. It can take long time because of some difficult procedures, but some general principles of good practice should not be neglected.
  9. I do not understand really the need to define what is the gold-standard between open and MIS in liver surgery. Some cases can be efficiently and safely done by MIS, but some others no. Being open liver surgery the rescue approach in MIS I suggest to the authors to remove sentence at rows 61-62.
  10. A paragraph named “study endpoints” should be added.
  11. Was this study registered in any independent international register for clinical trials? If yes, where and when?
  12. The study period is quite long. It would be important to know when these three types of procedures (OLR, LLR, RLR) were performed along the study period.
  13. The applied procedure for PSM should be better detailed. In particular, it is unclear to this reviewer the validity of three matched comparisons with different numbers (142 vs. 142; 22 vs. 22; 21 vs. 21); yet the groups of 22 and 21 patients are very limited. In this sense, it is a pity that from a large cohort of 1064 patients most were excluded in the final analysis.
  14. I wonder if more variables could be added in the analysis to better match the patients. In tables major resection is defined as the removal of more than 2 segments. This definition is in contrast with what the same authors declared before. Besides, the localization of the tumors should be considered as an important variable to be recorded and matched among groups. Were the lesions removed by the open approach in the same segments and with the same intrahepatic vascular relationships as those lesions removed by the lap and robotic approaches?
  15. Among variables, some explanations about liver cirrhosis should be given since CRLM usually do not have underlying cirrhosis.
  16. To me, a note of caution should be stated when reading about blood loss. First, blood loss is associated with the use of vascular control during the transection. Was the Pringle maneuver applied? Were the HV taped and clamped? Were these techniques applied in the matched groups? These data are missing. Second, blood loss as well as the use of Pringle maneuver might be considered as a proxy measures of complexity in liver surgery meaning that if we assume that the these different surgeons applied the same technique for the preparation (low CVP and other anesthesiology measures?!), transection and bleeding control, the increased in blood loss recorded in the open approach might be related to more complex tumor presentations.
  17. Assuming that these authors are very good liver surgeons, I assume that they can operate satisfying oncological criteria independently by the length of the skin wounds. Please note that the Kaplan Meier curves are biased by the fact these patients are not oncologically comparable: a lot of oncological data are missing (e.g. TNM of the primary, timing of CLM, mutational profiles, etc…). These KM curves give back the univariate comparison among groups with different tumor features. Back to the point number 16, if patients operated with the open approach had more complex tumor presentations but they had the same survival of those patients operated by the LLR or the RLR approaches, one could argue that in reality they had better survival.
  18. The authors should detail the complications. Which were these complications in the three approaches?
  19. The conversion rates are missing.
  20. Please consider refining the discussion which is difficult to follow.

Author Response

I have read with interest this manuscript that concerns the results of a multicentric retrospective study on open versus minimally invasive surgery for CRLM. While the topic might be of general interest to the readers of Cancers, I believe that there are some flaws that strongly limit the quality of this paper.

  1. The manuscript should be reviewed for proofreading and English grammar. There are also several typos throughout the paper and tables.

The entire manuscript was extensively edited and proofread.

  1. The title should be modified because it is unusual to start with “Study: … “. Similarly, the first sentence of the abstract should be modified.

The wording regarding the title was changed throughout the manuscript to make it more readable, we kept the use of SIMMILR because the main point of our paper was that preliminary overall and survival curves appear to be SIMILAR regardless of what approach is used.

The first line of the ABSTRACT now reads :

“This is a retrospective Study: of an International Multicentric cohort after Minimally Invasive Liver Resection (SIMMILR) of Colorectal Liver Metastases (CRLM) from 6 centers.“

The first line of the conclusion of the abstract now reads :

SIMMILR indicates that minimally invasive approaches for CRLM that follow the Milan Criteria may have short term advantages, but similar overall and recurrence free survival.

Additionally, the first line of the CONCLUSIONS section of the body of the manuscript now reads :

The Study: of International Multicentric Minimally Invasive Liver Resection  (SIMMILR) is a retrospective PSM study of CRLM, which reveals that LLR may result in decreased length of hospital stay and complication rates in the compared to OLR.

  1. In the abstract and in the main text the authors refer to the Milan Criteria as one of the inclusion criteria for this study. First, the authors should be better detail the rationale of this choice, meaning the exclusion of patients with more than 3 tumors measuring more than 5 cm. Second, such definition refers to transplantation criteria for HCC. This definition and concept should not be used for CRLM. Third, it is unclear the real application of this definition in this study since later in the text the authors report a case of surgery for a 17-cm single lesion, which should be excluded according with their CLM Milan Criteria.

This error was noted by several reviewers, thanks for picking this up.

The definition was clarified in the abstract :

“Patients with macrovascular invasion, ≥  3 tumors measuring more than 3cm or a solitary lesion  >5cm were excluded,”

The rationale for using the Milan Criteria and a more complete definition was added to the METHODS:

For the PSM we excluded patients with more than three tumors measuring ≥ 3cm, solitary lesions >5cm and evidence of microvascular invasion. Although originally devised to help in treating patients with hepatocellular cancer, the Milan Criteria have also been used to identify patients with CRLM that would benefit from liver resection, because of this we decided to use this as an exclusion criteria. This was also done to reduce the perceived bias that easier” tumors are removed minimally invasively when compared to open approaches.

This phrase was removed : “We performed propensity score matching (PSM) on only patients with ≤ 3 tumors with all tumors ≤ 5cm to adjust for potential cofounders and, thus, to reduce selection bias.”

This reference was also added : Chiba N, Abe Y, Koganezawa I, Nakagawa M, Yokozuka K, Ozawa Y, Kobayashi T, Sano T, Tomita K, Tsutsui R, Kawachi S. Efficacy of the Milan criteria as a prognostic factor in patients with colorectal liver metastases. Langenbecks Arch Surg. 2021 Jun;406(4):1129-1138.

  1. The introduction and the discussion include some paragraphs with specific citations that show a kind of enthusiastic way of writing that anticipates the results of the study. The paper should be written with more balance.

The INTRODUCTION was heavy edited and almost all of the case reports removed from the References.

  1. References: REF number 50 is missing; more importantly, there are approximately 20 references (31%) from the same authors. Self-citation should be limited to 15-20%. Besides, some of these citations are case reports, that might be omitted.

Reference 50 was added, thank you. Self-citation was reduced to < 20%.

  1. Most of the introduction concerns the history of published consensus conferences on MIS in liver surgery, which I believe is not pertinent and should be, if important to the authors, moved to the discussion.

Please see above. The INTRODUCTION was vastly edited with more than 2 entire paragraphs removed.

  1. In particular, the paragraph that goes from row 55 to row 62 is critical. I believe that the surgical approach should not change the indications in liver surgery. Never. The authors should change the word “indications” with something else. Otherwise, the risk is that – as already done in the past – MIS is used to remove asymptomatic small benign liver tumors that should not be touched (nor previously and for sure nor in 2022).

This section was removed.

  1. Along the point of benign tumors, I suggest to the authors to change the sentence at rows 315-317. It is unacceptable to justify surgery for benign liver tumors for training reasons. The training in surgery is not a race. It can take long time because of some difficult procedures, but some general principles of good practice should not be neglected.

This line was removed.

  1. I do not understand really the need to define what is the gold-standard between open and MIS in liver surgery. Some cases can be efficiently and safely done by MIS, but some others no. Being open liver surgery the rescue approach in MIS I suggest to the authors to remove sentence at rows 61-62.

This line was removed.

  1. A paragraph named “study endpoints” should be added.

This section was added :

“2.2. Study Endpoints

         We performed a retrospective review of all patients who underwent liver resection for colorectal metastases. For the PSM we excluded patients with more than three metastases measuring ≥ 3cm, solitary metastasiss >5cm and evidence of microvascular invasion. Although originally devised to help in treating patients with hepatocellular cancer, the Milan Criteria have also been used to identify patients with CRLM that would benefit from liver resection, because of this we decided to use this as an exclusion criteria[23]. This was also done to reduce the perceived bias that easier” tumors are removed minimally invasively when compared to open approaches. Patients who had undergone associating liver partition and portal vein ligation for staged hepatectomy (ALLPS), previous ablation and repeat liver resections were excluded. The files of patients started with either laparoscopy or robotic assistance were analyzed on an intention to treat basis. The data that support the findings of this study are available from the corresponding author upon reasonable request.

         For further comparative analysis patients were divided into three groups (OLR, LLR and RLR) depending on the surgical technique used for resection. The primary endpoint of the study was postoperative short-term mortality (death within 30 and 90 days) and overall survival as well as recurrence free survival. Secondary endpoints were intra-operative parameters (blood loss, OR time), length of hospital stay, complete oncologic resection and severe postoperative complications. The Dindo-Clavien classification system was used in case of postoperative complications with major complications defined as greater than grade 2[24]. Finally, propensity score matching was also done to get a more accurate comparison between techniques. Written informed consent was obtained on all patients. On the informed consent that is signed by each patient, it is explained that their anonymous patient data may be used to perform future studies. The STROBE statement checklist for reporting of observational studies was used during the drafting and editing of the manuscript.”

  1. Was this study registered in any independent international register for clinical trials? If yes, where and when?

This study was not registered with an independent international register, but the STROBE criteria were used to aid with reporting.

  1. The study period is quite long. It would be important to know when these three types of procedures (OLR, LLR, RLR) were performed along the study period.

Please see the new FIGURE 1 to see dates when the various techniques were begun during the study period.

  1. The applied procedure for PSM should be better detailed. In particular, it is unclear to this reviewer the validity of three matched comparisons with different numbers (142 vs. 142; 22 vs. 22; 21 vs. 21); yet the groups of 22 and 21 patients are very limited. In this sense, it is a pity that from a large cohort of 1064 patients most were excluded in the final analysis.

This was added to the METHODS to address this :

"We performed a retrospective review of all patients who underwent liver resection for colorectal metastases. For the PSM we excluded patients with more than three metastases measuring ≥ 3cm, solitary metastasiss >5cm and evidence of microvascular invasion. Although originally devised to help in treating patients with hepatocellular cancer, the Milan Criteria have also been used to identify patients with CRLM that would benefit from liver resection, because of this we decided to use this as an exclusion criteria [39]. This was also done to reduce the perceived bias that easier” tumors are removed minimally invasively when compared to open approaches. Patients who had undergone associating liver partition and portal vein ligation for staged hepatectomy (ALLPS), previous ablation and repeat liver resections were excluded. The files of patients started with either laparoscopy or robotic assistance were analyzed on an intention to treat basis. The data that support the findings of this study are available from the corresponding author upon reasonable request.”

“On the informed consent that is signed by each patient, it is explained that their anonymous patient data may be used to perform future studies. The STROBE statement checklist for reporting of observational studies was used during the drafting and editing of the manuscript.”

“Lesions in the Deep or Deeper segments were defined as metastases in segments 4B, 7or 8.”

“Statistical data analysis was performed using the Social Science Statistics software (www.socscistatistics.com) and SPSS (version 26; IBM, Armonk, New York, USA). Prism 8: GraphPad software (https://www.graphpad.com/scientific-software/prism/) was used to generate Kaplan-Meier curves, and the Log-rank (Mantel Cox) test was used to calculate p-values. OS was calculated from the date of liver resection. RFS was calculated from whichever date of diagnosis of recurrence was earliest. Recurrence was either diagnosed from rising serum tumor markers, radiographic examination, or upon positive histological confirmation from percutaneous or surgical biopsy.”

“Metastasis size was based on final pathological result.”

  1. I wonder if more variables could be added in the analysis to better match the patients. In tables major resection is defined as the removal of more than 2 segments. This definition is in contrast with what the same authors declared before. Besides, the localization of the tumors should be considered as an important variable to be recorded and matched among groups. Were the lesions removed by the open approach in the same segments and with the same intrahepatic vascular relationships as those lesions removed by the lap and robotic approaches?

The removal of 3 or more segments were indeed considered Major Resections, this inconsistency was corrected throughout the manuscript.

Furthermore, this was added to the METHODS :

         “We performed a retrospective review of all patients who underwent liver resection for colorectal metastases. For the PSM we excluded patients with more than three metastases measuring ≥ 3cm, solitary metastasiss >5cm and evidence of microvascular invasion. Although originally devised to help in treating patients with hepatocellular cancer, the Milan Criteria have also been used to identify patients with CRLM that would benefit from liver resection, because of this we decided to use this as an exclusion criteria [39]. This was also done to reduce the perceived bias that easier” tumors are removed minimally invasively when compared to open approaches. Patients who had undergone associating liver partition and portal vein ligation for staged hepatectomy (ALLPS), previous ablation and repeat liver resections were excluded.”

“If the Pringle was used even once, the technique was considered to have been utilized.”

“Lesions in the Deep or Deeper segments were defined as metastases in segments 4B, 7or 8.”

“…the Log-rank (Mantel Cox) test was used to calculate p-values. OS was calculated from the date of liver resection. RFS was calculated from whichever date of diagnosis of recurrence was earliest. Recurrence was either diagnosed from rising serum tumor markers, radiographic examination, or upon positive histological confirmation from percutaneous or surgical biopsy.”

“Metastasis size was based on final pathological result.”

An extensive discussion of Metastasis location in or out of the Deep segments and use of Pringle maneuver was added to the RESULTS section and Tables.

RESULTS

         “Lastly, lesions were significantly more prevalent in the deeper segments in the OLR group, p= 0.03. No differences were noted in the utilization of the Pringle maneuver.

When unmatched patients who had undergone OLR were compared to RLR, ASA class and presence of cirrhosis was significantly higher in the RLR group, p= 0.02 and 0.03, respectively), however, significantly more patients received neoadjuvant chemotherapy, had undergone previous surgery and had on average higher pre-oerative CEA serum levels, p= 0.03, 0.04 and 0.08, respectively. Furthermore more metastases that were larger were removed in the OLR group, p= 0.003 and 0.007, respectively. Although lesions tended to be in the deeper segments more often in the OLR group, this did not attain statistical significance, p=0.06. The Pringle maneuver was used significantly more frequently in the OLR group when compared to the RLR group, p= 0.004)

         When unmatched patients who had undergone LLR were compared to RLR, the ASA class was statistically higher in the RLR group, p= 0.001, and the Pringle maneuver was used statistically more often in the LLR group, p=0.02.”

3.2. Differences between open (OLR) and robotic liver resection (RLR)

“…, location in the Deep segments or utilization of the Pringle maneuver between the 2 approaches. Although before matching patients in the RLR group had a significantly higher prevalence of liver cirrhosis (10.7 % vs. 0.3 %; p<0.001) and severe co-morbidities indicated by significantly higher mean ASA score (2.5 vs. 2.1; p<0.001); and metastases were found significantly more often in the Deep segments (44.0% vs. 35.7%; p=0.04) and the Pringle maneuver was used more frequently in the OLR group (25% vs. 3.6%; p=0.009),…”

“3.3. Differences between laparoscopic (LLR) and robotic liver resection (RLR)

         Table 4 describes demographics and clinical outcome variables of LLR and RLR. Although before matching patients in the RLR group had a significantly higher number of metastases resected (1.4 vs.1.0; p=0.003); and significantly more major resections (47.5% vs. 17.9%; p=0.002) and the Pringle maneuver  (28.0% vs. 3.6%; p=0.003)was used more frequently in the LLR group, these differences were insignificant after the matching process. “

  1. Among variables, some explanations about liver cirrhosis should be given since CRLM usually do not have underlying cirrhosis.

This was added to the RESULTS :

Cirrhosis was due to either steatohepatitis or alcohol abuse. No patients had viral hepatitis and all cirrhotics were Child’s A.

  1. To me, a note of caution should be stated when reading about blood loss. First, blood loss is associated with the use of vascular control during the transection. Was the Pringle maneuver applied? Were the HV taped and clamped? Were these techniques applied in the matched groups? These data are missing. Second, blood loss as well as the use of Pringle maneuver might be considered as a proxy measures of complexity in liver surgery meaning that if we assume that the these different surgeons applied the same technique for the preparation (low CVP and other anesthesiology measures?!), transection and bleeding control, the increased in blood loss recorded in the open approach might be related to more complex tumor presentations.

This was addressed above in the Methods, Tables and Results. This was added to the end of the DISCUSSION to help further clarify this.

         “Although there is significant heterogeneity between the approaches used by center, the centers included in this study are referral centers and are highly experienced in minimally invasive surgery, as a result, the presented data might not be totally representative for real world situations. Another limitation of the study is the lack of longer follow-up and low numbers, particularly, in the robotic group. Additionally, the case number of RLR is small and the cases are from only 3 centers. This is due to the fact that robotic surgery of the liver is still in its infancy. We had strict criteria inclusion and exclusion criteria for our PSM and needed to include non-robotic centers to get enough patients to make any meaningful comments on the various approaches to liver surgery. Before matching, Pringle maneuver was used significantly more during OLR and LLR when compared to RLR, after matching this difference was no longer statistically significant. The observation that patients had on average 540mL more blood loss after LLR and RLR after matching is probably due to the fact that more patients had tended to have cirrhosis in the LLR arm.

         In the future, as RLR becomes more common we believe that it will indeed be possible to do a better multicentric study where each center has enough open, laparoscopic and robotic liver resection, however, we are not there yet. Importantly, we believe that future liver surgeons should become versed in all 3 approaches and that future studies on liver surgery should include data from all 3 techniques and not solely be limited to binary analyses (i.e. OLR vs RLR).

  1. Assuming that these authors are very good liver surgeons, I assume that they can operate satisfying oncological criteria independently by the length of the skin wounds. Please note that the Kaplan Meier curves are biased by the fact these patients are not oncologically comparable: a lot of oncological data are missing (e.g. TNM of the primary, timing of CLM, mutational profiles, etc…). These KM curves give back the univariate comparison among groups with different tumor features. Back to the point number 16, if patients operated with the open approach had more complex tumor presentations but they had the same survival of those patients operated by the LLR or the RLR approaches, one could argue that in reality they had better survival.

Please see above. We believe that by adding information on Metastasis location, use of Pringle maneuver and clarification of the definition and rationale for why the Milan Criteria that a justification for reporting our Kaplan Meier curves is more reasonable.

Please again refer to the METHODS section :

“We performed a retrospective review of all patients who underwent liver resection for colorectal metastases. For the PSM we excluded patients with more than three metastases measuring ≥ 3cm, solitary metastasiss >5cm and evidence of microvascular invasion. Although originally devised to help in treating patients with hepatocellular cancer, the Milan Criteria have also been used to identify patients with CRLM that would benefit from liver resection, because of this we decided to use this as an exclusion criteria [39]. This was also done to reduce the perceived bias that easier” tumors are removed minimally invasively when compared to open approaches. Patients who had undergone associating liver partition and portal vein ligation for staged hepatectomy (ALLPS), previous ablation and repeat liver resections were excluded.”

  1. The authors should detail the complications. Which were these complications in the three approaches?

After matching no mortalities were noted in the patients with CRLM that followed the Milan Criteria. All major complications involved either percutaneous drainage of bile leaks or abscesses and/or need for endoscopic biliary drainage. No re-operations were noted.

This information was added to each RESULTS Section (OLR vs LLR; OLR RLR and LLR vs. LLR, respectively).

This was indicated in each section.

  1. The conversion rates are missing.

The conversion rates are listed in the Tables.

  1. Please consider refining the discussion which is difficult to follow.

The DISCUSSION and CONCLUSION have both been edited heavily to address the issues raised.

Round 2

Reviewer 1 Report

The authors carefully adopted the manuscript to the reviewer´s comments. Please correct minor errors below:

Revised manuscript - Methods - page 3:

"Methods  For the PSM we excluded patients with more than three metastases measuring ≥ 3cm, solitary metastasiss >5cm and evidence of microvascular invasion."

--> please correct  "metastasiss" and "micovasular" to "macrovascular"

Results - page 4:

After matching, the percentage of liver resections done via an open, laparoscopic or robotic approach by center was :
45.4/69.3/0 (center 1), 36.8/3.1/39.6 (center 3), 0/12.9%/0 (center 4), 6.1/1.8/30.2 (center 5) and 11.7/12.9/30.2 (center 6), respectively. 

--> please remove superfluos percent sign

Author Response

We have incorporated the requested changes. Thank you for reviewing our manuscript.

Our kind regards,

Professor Andrew A. Gumbs MD, FACS Editor-in-Chief of Artificial Intelligence Surgery (aisjournal.net) Chirurgien Hépato-pancreato-biliaire Responsable de la filière cancérologie digestive, GHT-Yvelines, Nord, Yvelines-Nord Région Département de Chirurgie Viscéral Centre Hospitalier Intercommunal de Poissy/Saint-Germain-en-Laye Poissy 78300 France   Grigol Robakidze University Tblisi, 0159 Georgia   Department of Surgery Otto von Guericke University Magdeburg Magdeburg, 39120 Germany

Reviewer 3 Report

I think that this manuscript is acceptable in this form.

Author Response

Thank you very much for your extremely useful comments! We believe that the manuscript has been greatly improved. And also, thank you for accepting our manuscript.

Our kind regards,

Professor Andrew A. Gumbs MD, FACS Editor-in-Chief of Artificial Intelligence Surgery (aisjournal.net) Chirurgien Hépato-pancreato-biliaire Responsable de la filière cancérologie digestive, GHT-Yvelines, Nord, Yvelines-Nord Région Département de Chirurgie Viscéral Centre Hospitalier Intercommunal de Poissy/Saint-Germain-en-Laye Poissy 78300 France   Grigol Robakidze University Tblisi, 0159 Georgia   Department of Surgery Otto von Guericke University Magdeburg Magdeburg, 39120 Germany

Reviewer 4 Report

It is an interesting paper with important information. The authors did good revision for the reviewer's comments. I think this manuscript is suitable for the publication in the journal.

Author Response

This is great news. Thank you VERY much.

My kind regards,

Professor Andrew A. Gumbs MD, FACS Editor-in-Chief of Artificial Intelligence Surgery (aisjournal.net) Chirurgien Hépato-pancreato-biliaire Responsable de la filière cancérologie digestive, GHT-Yvelines, Nord, Yvelines-Nord Région Département de Chirurgie Viscéral Centre Hospitalier Intercommunal de Poissy/Saint-Germain-en-Laye Poissy 78300 France   Grigol Robakidze University Tblisi, 0159 Georgia   Department of Surgery Otto von Guericke University Magdeburg Magdeburg, 39120 Germany

Reviewer 5 Report

I have read again this paper and the authors’ replies. While I thank the authors for their efforts, I believe that this manuscript has some major flaws due the selection and aggregation biases. Please be advised that your KM curves are still biased because your patients are not oncologically comparable. This study was not done to investigate survival otherwise you should have recorded much more data (e.g. TNM of the primary, timing of CLM, chemotherapy, mutational profiles, etc…). The add of Pringle maneuver or CLM locations does not help too much since these data are not relevant in terms of long-term prognosis.

Please consider to remove the survival analyses.

Author Response

The tumor location and Pringle maneuver data was requested by several of the other Reviewers.

We understand your remaining concerns, but still feel that the survival curve data is valuable. Furthermore, the four other reviewers have accepted the manuscript so we feel that eliminating the curves would not do their decisions justice. Future studies will be prospective and will incorporate your recommendations.

Our kind regards,

Professor Andrew A. Gumbs MD, FACS Editor-in-Chief of Artificial Intelligence Surgery (aisjournal.net) Chirurgien Hépato-pancreato-biliaire Responsable de la filière cancérologie digestive, GHT-Yvelines, Nord, Yvelines-Nord Région Département de Chirurgie Viscéral Centre Hospitalier Intercommunal de Poissy/Saint-Germain-en-Laye Poissy 78300 France   Grigol Robakidze University Tblisi, 0159 Georgia   Department of Surgery Otto von Guericke University Magdeburg Magdeburg, 39120 Germany

Round 3

Reviewer 5 Report

Dear authors, at this second round of revision you did not provide the suggested changes regarding the survival analyses. Since you did not provide enough evidence that these groups were balanced, at least this part of the results should be revised/deleted. This remains as a major flaw that strongly limit the scientific soundness of your interesting study.

Author Response

We have deleted all of the survival data as requested.

Please see the edited manuscript.